# Clonal replacement and heterogeneity in breast tumors treated with neoadjuvant HER2-targeted therapy

Jennifer L. Caswell-Jin [1], Katherine McNamara [1,2,3], Johannes G. Reiter [4], Ruping Sun [1,2,3], Zheng Hu[1,2,3], Zhicheng Ma[1,2,3], Jie Ding[1,2,3], Carlos J. Suarez[5], Susanne Tilk[6], Akshara Raghavendra[7], Victoria Forte[8,9], Suet-Feung Chin [10], Helen Bardwell[10], Elena Provenzano[11], Carlos Caldas [10], Julie Lang [9,12], Robert West[5], Debu Tripathy [7], Michael F. Press[9,13] & Christina Curtis [1,2,3]

Genomic changes observed across treatment may result from either clonal evolution or geographically disparate sampling of heterogeneous tumors. Here we use computational modeling based on analysis of fifteen primary breast tumors and find that apparent clonal change between two tumor samples can frequently be explained by pre-treatment heterogeneity, such that at least two regions are necessary to detect treatment-induced clonal shifts. To assess for clonal replacement, we devise a summary statistic based on whole-exome sequencing of a pre-treatment biopsy and multi-region sampling of the post-treatment surgical specimen and apply this measure to five breast tumors treated with neoadjuvant HER2-targeted therapy. Two tumors underwent clonal replacement with treatment, and mathematical modeling indicates these two tumors had resistant subclones prior to treatment and rates of resistance-related genomic changes that were substantially larger than previous estimates. Our results provide a needed framework to incorporate primary tumor heterogeneity in investigating the evolution of resistance.

[1] Department of Medicine, Division of Oncology, Stanford University School of Medicine, Stanford 94305 California, United States. [2] Department of Genetics, Stanford University School of Medicine, Stanford 94305 CA, USA. [3] Stanford Cancer Institute, Stanford University School of Medicine, Stanford 94305 CA, USA. [4] Canary Center for Cancer Early Detection, Department of Radiology, Stanford University School of Medicine, Palo Alto 94305 CA, USA. [5] Department of Pathology, Stanford University School of Medicine, Stanford 94305 CA, USA. [6] Department of Biology, Stanford University, Stanford 94305 CA, USA. [7] Department of Breast Medical Oncology, The University of Texas MD Anderson Cancer Center, Houston 77030 TX, USA. [8] Maimonides Medical Center, Brooklyn 11219 NY, USA. [9] Norris Comprehensive Cancer Center, Los Angeles 90033 CA, USA. [10] Cancer Research UK Cambridge Institute, Department of Oncology, University of Cambridge, Cambridge CB2 0RE, UK. [11] Cambridge Experimental Cancer Medicine Centre and NIHR Cambridge Biomedical Research Centre, Cambridge University Hospitals NHS Foundation Trust, Cambridge CB2 0QQ, UK. [12] Department of Surgery, Keck School of Medicine, University of Southern California, Los Angeles 90333 CA, USA. [13] Department of Pathology, Keck School of Medicine, University of Southern California, Los Angeles 90033 CA, USA. These authors contributed equally: Jennifer L. Caswell-Jin, Katherine McNamara. Correspondence and requests for materials should be addressed to C.C. (email: cncurtis@stanford.edu)

Breast tumors exhibit cell-to-cell, spatial, and temporal heterogeneity, with important implications for the use of targeted therapies[1–4]. Several studies have suggested that tumors may change substantially from diagnosis to surgery over the course of only a few months of neoadjuvant (pre-operative) therapy[4–8]. The three established breast cancer expression biomarkers—estrogen receptor (ER), progesterone receptor (PR), and HER2—are discordant between the primary tumor and the residual tumor after neoadjuvant therapy in 5–40% of cases that do not achieve a pathologic complete response (pCR, defined as absence of remaining invasive cancer)[5,6]. Sequencing of triple-negative[4,8] and ER-positive[7] breast cancers has suggested that breast tumors can undergo major shifts in genomic aberrations with neoadjuvant therapy. These results raise the possibility that treatment-induced selective pressure may lead to the outgrowth of resistant subclones undetected in the pre-treatment tumor[9]. Such an evolutionary model would have implications for treatment, including whether adjuvant therapies should be targeted based on the genetic composition of the post-treatment rather than the pre-treatment tumor[10,11].

Importantly, however, it is not clear whether and when observed tumor changes with neoadjuvant therapy reflect treatment-induced clonal evolution rather than pre-existing intra-tumor heterogeneity (ITH). Existing sequencing studies typically use a single sample from the pre- and post-treatment time points[4,8], but, based on targeted sequencing panels, primary breast tumors may exhibit substantial spatial ITH at a single time point[2]. Given the possibility of extensive pre-existing ITH within the tumor, it may not be readily apparent whether a genomic shift across time reflects geographically disparate sampling or a true evolutionary change in the tumor's composition. In order to understand how breast tumors evolve with therapy, an analytical framework that considers ITH is needed.

Early-stage HER2-positive breast cancers afford the optimal setting to study genomic changes in breast tumors treated with targeted combination therapy. Many stage II–III HER2-positive breast cancers are treated with neoadjuvant HER2-targeted therapy combined with chemotherapy, with improved rates of response at time of surgery compared to chemotherapy alone[12]. Patients with residual disease after neoadjuvant therapy tend to have worse outcomes than those who obtain a pCR[13]. However, the value of genomic profiling of residual disease in these patients —to identify causes of resistance or predict effect of future treatments—remains unclear.

To assess and account for baseline primary breast ITH, we analyzed multi-region whole-exome sequencing data from 15 untreated primary breast tumors across clinical subtypes. We then performed whole-exome sequencing on a separate, non-overlapping cohort of a single pre-treatment diagnostic core biopsy and multiple regions of the post-treatment surgical specimen from five archival HER2-positive breast tumors that were treated with neoadjuvant HER2-targeted therapy combined with chemotherapy and did not achieve a pCR. We use these data to determine evolutionary trajectories and mathematical modeling to define evolutionary parameters that could lead to these trajectories. The picture we uncover is one of high and markedly variable heterogeneity that is present in both untreated and treated primary breast tumors, even as a subset of tumors undergo vast shifts in clonal architecture with treatment. We also infer wide variability in evolutionary parameters across breast tumors, including the rate of mutation and copy number change as well as the number of available paths to resistance, that dictate in large part the outcomes of neoadjuvant therapy.

## Results

**Untreated breast tumors exhibit high spatial heterogeneity.** To understand the potential impact of ITH in interpreting treatment-induced genomic change, we first assessed ITH in untreated primary breast tumors. We analyzed whole-exome sequencing data from four multi-region sampled primary breast tumors generated for this study, as well as data from 11 tumors from four previous studies that performed genome or exome sequencing of high-quality multi-region samples from primary breast tumors[2,14–16] (Supplementary Data).

We quantified regional ITH in these 15 primary breast tumors using two complementary statistics: Wright's fixation index (Fst) (Eq. (1))[17,18] and high-frequency regional (HFR) (Eq. (2)), defined here as the percentage of high-frequency mutations (cancer cell fraction (CCF) > 0.5) from a single region that are absent or rare (CCF < 0.1) in a second, spatially disparate region (Supplementary Fig. 1a). HFR uses mutations with drastic, and therefore reliable, differences in CCF between two samples and is thus robust to moderate differences in purity and sequencing coverage between samples[19]. Conversely, Fst uses subclonal mutations and is independent of the number of truncal mutations, many of which occurred prior to transformation. While Fst therefore informs the tumor's post-transformation evolutionary history, HFR provides direct information on the clinical utility of multi-region sampling, corresponding to the percentage of possibly clonal mutations found to be subclonal when analyzing a second region of the tumor (Methods).

Both Fst and HFR were highly variable across the 15 primary breast tumors, with some tumors with very low heterogeneity and others with higher levels of heterogeneity than has been seen in other tumor types[17,20–25] (Fig. 1a, b). We did not observe significant differences between breast cancer clinical subtypes, though numbers within each subtype were limited ($n = 6$ ER-positive/HER2-negative, $n = 2$ HER2-positive, and $n = 7$ triple-negative). Mean HFR was 26% (range 1–70%), and was significantly higher in breast tumors than other tumor types (Fig. 1b). Breast tumors also had modestly lower purity (62% vs 78%, two-sided $t$-test $p = 2.6 \times 10^{-4}$) and coverage (90× vs 115×, two-sided $t$-test $p = 0.05$) than other tumor types, but in a multivariate model including purity, coverage, and tumor type (breast vs other), only tumor type remained significant (multivariate $t$-test $p = 0.015$ vs $p = 8.5 \times 10^{-5}$ in univariate analysis). Mutations in the canonical breast cancer drivers PIK3CA and TP53 were clonal in all regions when present ($n = 11$). However, other drivers[26] or putative targets of therapy[27] could be either clonal in all regions ($n = 14$), subclonal in one or more regions ($n = 29$), or clonal in one region but absent in another (HFR) ($n = 8$) (Supplementary Data).

We next adapted our previously described spatial tumor growth models[17,20] to simulate thousands of realistically sized virtual tumors (composed of $10^9$ cells) in order to infer evolutionary parameters that could lead to the patterns of ITH observed in the primary breast tumors (Methods). Higher ITH as measured by mean HFR resulted both from tumor growth under selection ($s \geq 0.05$, where $s$ corresponded to the change in the cell's growth rate when an advantageous mutation arose, with $s = 0$ reflecting neutral evolution) (Fig. 1c) and under greater spatial constraints (smaller deme sizes, with a deme corresponding to a well-mixed tumor cell subpopulation, such as a neoplastic gland) (Supplementary Fig. 3). Fourteen of 15 primary breast tumors had ITH levels inconsistent with neutral growth when compared with simulated tumors using a statistical inference framework based on Approximate Bayesian Computation (ABC) (Supplementary Fig. 2). The high variability in ITH that was observed in the patient tumors was also recapitulated by the virtual tumors that grew under selection (Fig. 1c).

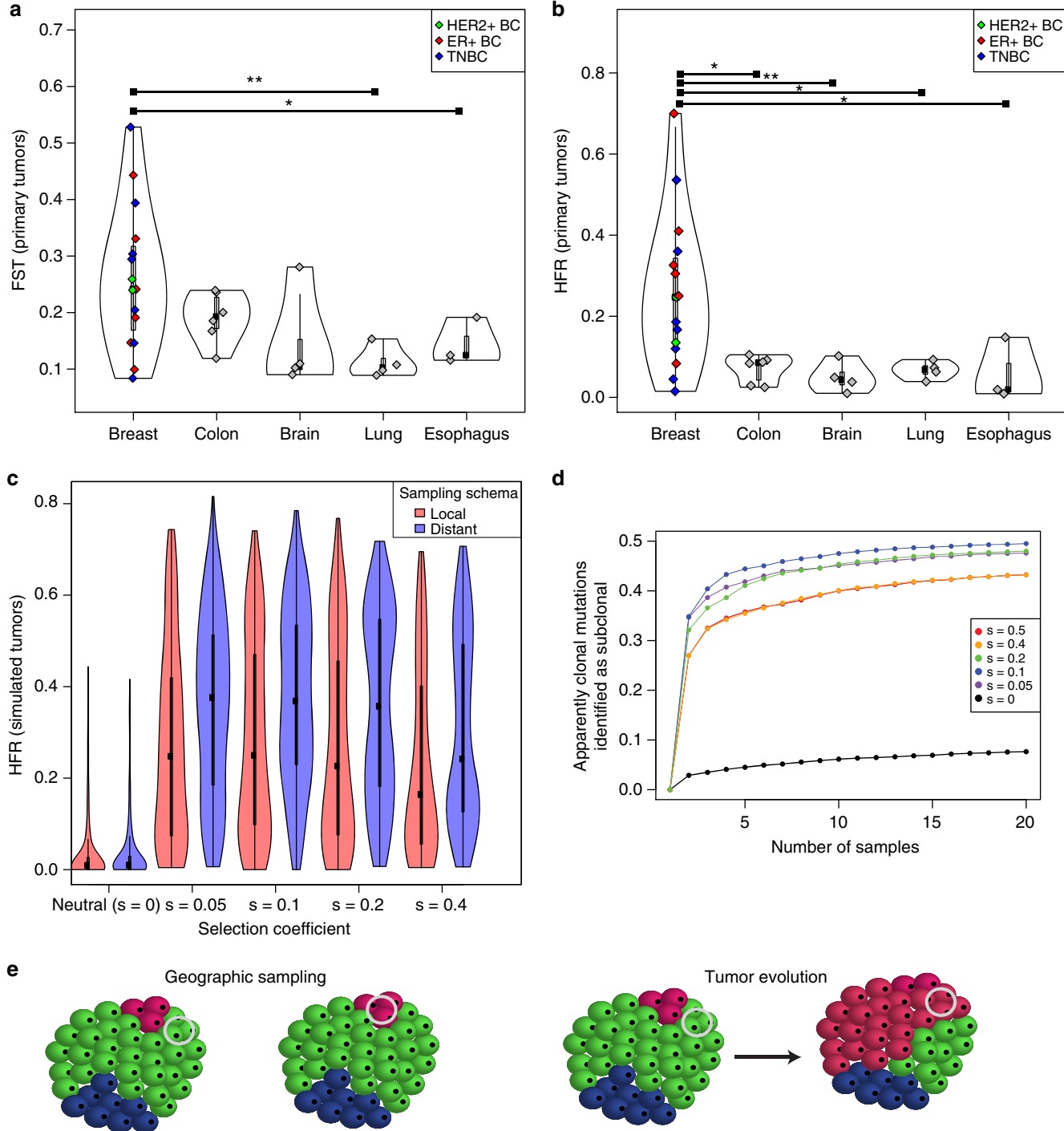

**Fig. 1** Intra-tumor heterogeneity of untreated primary breast tumors. **a** Average pairwise Fst between regions in fifteen breast tumors vs other untreated primary tumors: colon adenocarcinoma[20], glioma/gliobastoma (brain)[21,22], lung adenocarcinoma[23,24], and esophageal adenocarcinoma[25]. *$p < 0.05$, **$p < 0.01$ (two-sided $t$-test). **b** Average pairwise HFR (high-frequency regional) in breast vs other tumors. HFR corresponds to the percentage of high-frequency mutations from one region that are absent or rare in a second region. **c** HFR across tumors simulated with varying degrees of selection conferred by driver mutations. Local samples were taken from the same octant of the virtual tumor and distant samples were from distinct octants. **d** In virtual tumors, the mean proportion of apparently clonal (cancer cell fraction > 0.5) mutations in one region that were reclassified as definitively subclonal (cancer cell fraction < 0.1) upon sampling 1–19 additional regions at random, with varying degrees of selection. **e** Schematic demonstrating how, in tumors with high intra-tumor heterogeneity, geographic sampling may mimic tumor evolution. The different colored tumor cells represent distinct clones within the tumor and the silver circles indicate sampled regions. BC breast cancer, TN triple-negative. Source data for panels **a**–**d** are provided as a Source Data file

We next used the virtual tumors to assess the impact of different sampling approaches on classifying mutations as clonal or subclonal, a task with implications for precision oncology. In tumors that grew under neutral evolution, a second region rarely provided additional information, whether that region came from within the same octant (local sampling) or from a different octant (distant sampling) (mean HFR = 0.03 for both sampling schema). In tumors that grew under high levels of selection, a second biopsied region frequently showed clonal differences from the first region, and these differences were modestly greater with distant sampling (mean HFR = 0.35) than with local sampling (mean HFR = 0.27) (two-sided $t$-test $p < 1E$-15) (Fig. 1c). We similarly used the virtual tumors to assess the value of obtaining additional samples in mutation classification. In tumors that grew under selection, on average 31% of the mutations identified as having CCF > 0.5 in a first region were identified to be definitively subclonal (CCF < 0.1 in other regions) with one additional region and 36% with two additional regions. Continued diminishing gain in reclassification up to 47% was observed with 20 regions (Fig. 1d, Supplementary Fig. 4). In the five untreated patient tumors with sequencing data from more than two regions, we again saw a substantial increase in the percentage of clonal mutations from the first region that were rare in a second, and a more modest increase with additional regions (Supplementary Table 1). In other words, in both virtual and patient tumors, while substantial information about the clonal status of mutations was gained from obtaining two regions from a tumor, diminishing gain was obtained with three or more regions.

The findings from both the patient and virtual tumors suggested that if two biopsies are obtained randomly from a primary breast tumor at diagnosis, more than a quarter of the clonal mutations observed in one biopsy would be absent in the other simply due to the ITH present prior to treatment. In other words, the high level of ITH in breast tumors has substantial implications for understanding changes in tumor composition over time, including with therapy, as geographic genetic differences can mimic genetic divergence observed under clonal evolution (Fig. 1e).

**Heterogeneity is high in bulky tumors after treatment.** We next turned to treated tumors to understand how heterogeneity may change with neoadjuvant therapy. We began with a set of 20 archival stage II–III primary HER2-positive breast tumors that were treated with neoadjuvant chemotherapy combined with HER2-targeted therapy: for each, we performed whole-exome sequencing on one pre-treatment region (from the diagnostic core needle biopsy), one post-treatment region (from the surgical specimen), and matched normal tissue (from the surgical specimen). Approximately half (45%) of the initial cohort had cellularity at time of surgery too low to allow interpretable bulk whole-exome sequencing (Supplementary Fig. 5, Methods). As expected, excluded tumors tended to be smaller at time of surgery than those with adequate cellularity for downstream analysis (Supplementary Table 2). In the five tumors with adequate post-treatment cellularity/sequencing coverage and available tissue for multi-region sampling, we sequenced an additional 1–5 regions from separate pathology blocks of the surgical specimen (Supplementary Data). The tumor information, treatment courses, and sampling schema are provided in Fig. 2.

Among these tumors with relatively bulky residual disease, HFR was similar between multi-region sampled pre- (26%, range 1–70%) and post- (28%, range 10–54%) treatment tumors (Supplementary Fig. 6). As in pre-treatment tumors, mutations in the canonical breast cancer driver genes, *PIK3CA* and *TP53*, were generally clonal within each region pre- and post-treatment

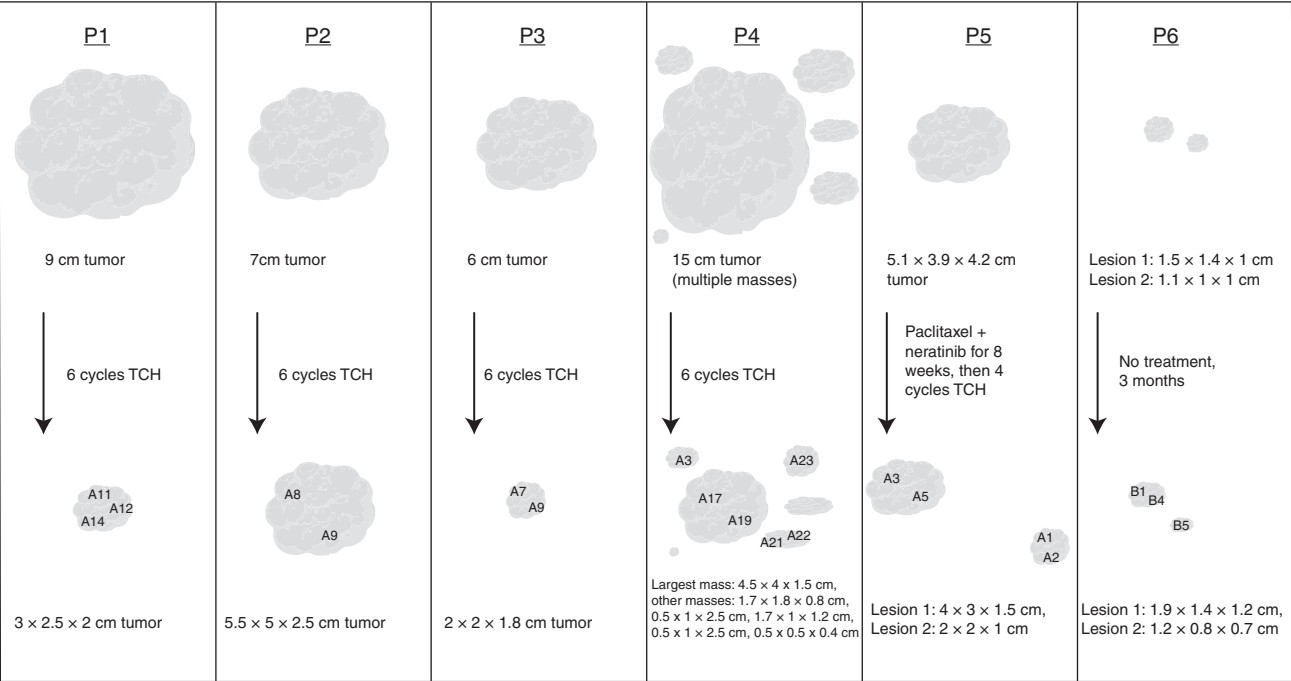

**Fig. 2** Longitudinal multi-region HER2-positive breast cancer cohort. Pre-treatment tumor sizes were determined from the medical oncologist's initial measurements. Post-treatment (or in P6, surgical) tumor sizes were determined from the pathologist's report of the surgical specimen. Tumors are drawn to scale and approximate locations of the tumor blocks from which the multi-region samples are shown. *TCH* docetaxel/carboplatin/trastuzumab. Tumor shapes from BioRender

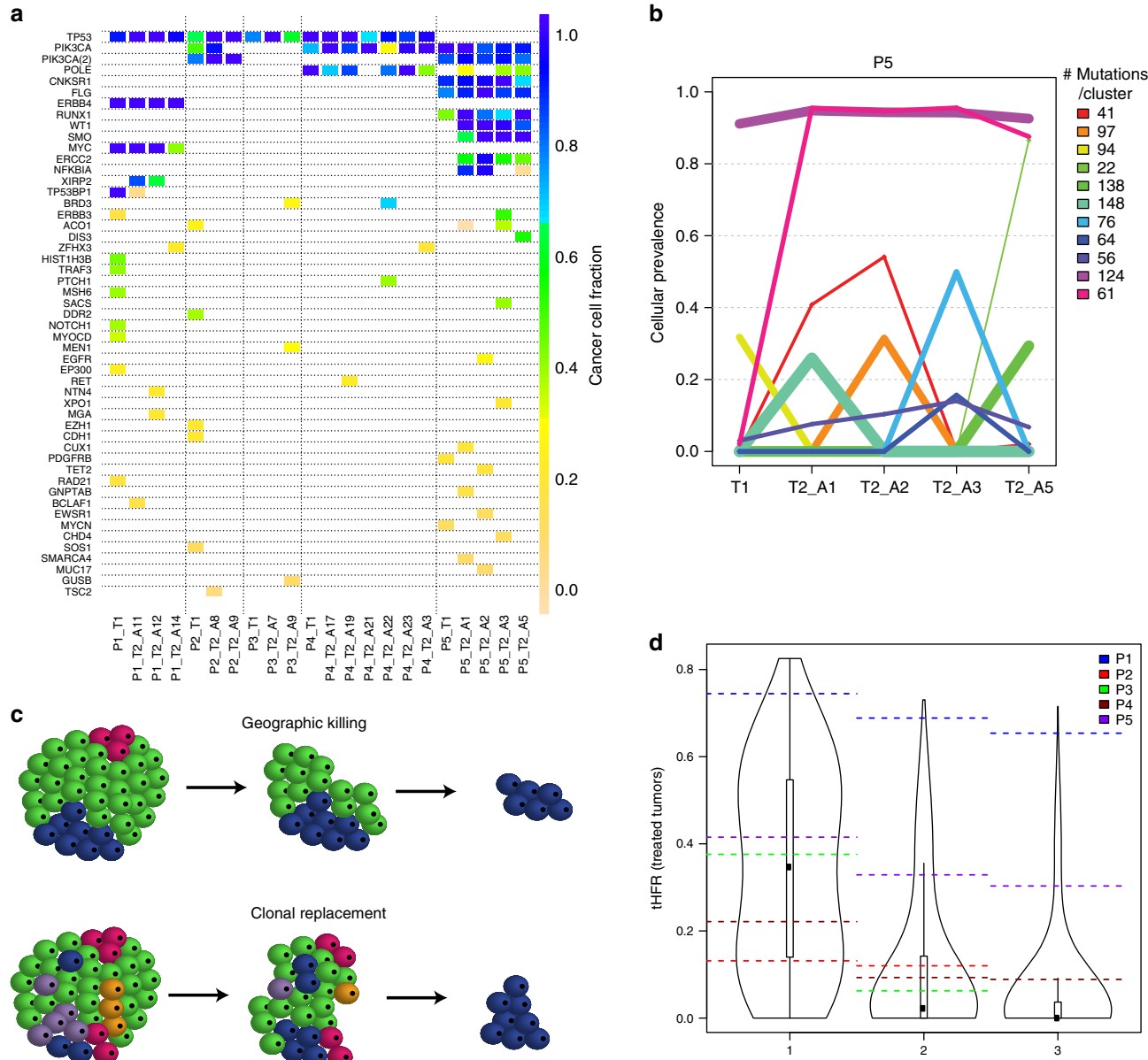

**Fig. 3** Intra-tumor heterogeneity and clonal replacement in treated primary breast tumors. **a** Heterogeneity of protein-altering mutations in driver and targetable genes pre- and post-treatment, with multi-region sampling. **b** Cellular prevalence of inferred mutational clusters across treatment in P5 (PyClone). T1 is pre-treatment and T2 is post-treatment. The purple cluster reflects truncal mutations and the pink cluster reflects a new clone arising post-treatment. **c** Schematic describing geographic killing and clonal replacement. Untreated primaries must have regions of high homogeneity for geographic killing to mimic clonal replacement. **d** Values of tHFR in patient-treated tumors (dashed lines) compared to distributions of tHFR in virtual untreated tumors. For each case, tHFR is computed using 1, 2, and, when possible, 3 post-treatment samples (averaged over all possible combinations of subsets of post-treatment samples) to compare to the distribution derived from the virtual tumors for 1, 2, and 3 post-treatment samples. tHFR corresponds to the percentage of clonal mutations across all post-treatment samples that are absent or rare in a single pre-treatment sample. P1 and P5 exhibited tHFR values consistent with clonal evolution (not geographic killing). In certain cases, at least two post-treatment samples were necessary to discriminate between pre-treatment heterogeneity (P3) and clonal evolution (P5), as demonstrated within the simulation framework. Source data for panels **a**, **b**, and **d** are provided as a Source Data file

(Fig. 3A, Supplementary Data). However, one tumor had two distinct *PIK3CA* mutations with one absent in one region and one tumor had a *TP53* deletion event present in all post-treatment regions that was absent pre-treatment (Supplementary Data). ITH in other driver[26] and putatively targetable[27] protein-altering mutations was high: within each of the five post-treatment tumors, 50–75% of the protein-altering driver or targetable mutations present in any post-treatment region were found in only one region (Fig. 3A, Supplementary Data). Across all five

tumors, the majority of region-specific driver or targetable mutations (n = 26) were present at low frequency, but four were high frequency (CCF > 0.5) and thus used to compute HFR.

**Clonal replacement can occur with neoadjuvant treatment.** Despite the generally high HFR between pairs of regions of post-treatment samples, two of the five tumors (P1 and P5) had a cluster of mutations that was clonal (CCF > 0.5) in all post-

treatment regions and absent or rare (CCF < 0.1) in the pre-treatment region (Fig. 3B, Supplementary Fig. 7). In theory, tumors with high heterogeneity between distant regions, but low heterogeneity within local regions, could generate such a cluster by geographic killing rather than treatment-induced clonal evolution (Fig. 3c); in such a scenario, the residual post-treatment tumor would represent a homogeneous area that had been present in the pre-treatment tumor. It was not feasible to sample the primary breast tumors with adequate granularity to determine if there were areas of high homogeneity within a heterogeneous tumor, especially recognizing the great variability seen in both the patient and virtual tumors. We therefore turned to the virtual untreated tumors to assess geographic killing as a possible explanation for the observed clonal shifts, as these recapitulated primary breast tumor heterogeneity (Fig. 1b, c). For each virtual tumor, we took one region from an octant as the pre-treatment sample and two regions from the octant on the opposite side of the tumor as the post-treatment samples (Supplementary Fig. 8a). We then defined a new statistic temporal high-frequency regional (tHFR) (Eq. (3)) to represent degree of clonal change across time much as HFR represents degree of clonal change across space (Supplementary Fig. 1b): tHFR here was equal to the proportion of mutations with CCF > 0.5 in all post-treatment regions that were absent or rare (CCF < 0.1) in the pre-treatment region of the tumor. This analysis confirmed that the clonal cluster seen post-treatment in P1 and P5 was statistically unlikely to have resulted from pre-existing heterogeneity in the untreated tumor (P1: tHFR = 0.65, >99.9th percentile of virtual tumors; P5: tHFR = 0.30, >90th percentile, both tHFR values computed with the maximal number of post-treatment samples available) (Fig. 3d). In other words, geographic killing alone was unlikely to explain the degree of clonal change observed across treatment, and so treatment-induced clonal evolution was likely to have contributed at least in part. The other three cases, as expected, had tHFR values consistent with heterogeneity in the untreated tumor (P2: 0.12, P3: 0.06, and P4: 0.09). Importantly, while the drastic clonal change across treatment was suggested with only one post-treatment region for P1 (>98th percentile of virtual tumors), multiple regions were needed to discriminate between pre-treatment heterogeneity (P3) or treatment-induced change (P5) for other tumors (Fig. 3d).

Notably, when the sampling scheme is known more precisely, it is possible to sample from the virtual tumors in accordance with that scheme. In particular, in P5, it was known that two of each of the four post-treatment regions came from opposite quadrants of the tumor (Fig. 2). We therefore generated a distribution of tHFR values in accordance with this sampling scheme (Supplementary Fig. 8B) and observed that the degree of clonal change seen in P5 was statistically unlikely to have resulted from pre-existing heterogeneity (tHFR > 99th percentile) (Supplementary Fig. 9). Thus details of the tissue sampling scheme, even if somewhat coarse, can be valuable in assessing for clonal evolution.

We next assessed if the clonal similarity seen among post-treatment samples in P1 and P5, as compared to the pre-treatment sample, was detectable using copy number aberrations rather than mutations (Supplementary Fig. 10). For this purpose, we calculated the copy number distance[28] between each pair of regions in the tumor, within and across timepoints. In the three tumors that, based on mutational data, did not appear to have undergone clonal replacement (P2, P3, and P4), the copy number distance between pairs of post-treatment samples and between pre- and post-treatment samples was similar, indicative of ongoing regional copy number heterogeneity. Conversely, in the two tumors that appeared to have undergone clonal replacement (P1 and P5), there was a trend toward a smaller copy number distance between post-treatment samples than between pre- and

post-treatment samples (two-sided t-test $p = 0.06$ for both P1 and P5).

While we only identified two clones consistent with clonal replacement (in P1 and P5), the genetic alterations contained within them represent candidate drivers of resistance, and are provided (Supplementary Data).

**Evolutionary trajectories during neoadjuvant treatment.** We estimated the range of possible pre-treatment resistant clone sizes that could lead to clonal replacement over the course of neoadjuvant therapy in P1 and P5. Assuming exponential growth at rates consistent with the literature on breast tumors[29–31], the resistant cell populations for both tumors made up 0.02–12.5% of the overall pre-treatment cell population (Fig. 4a). Notably, these estimates of resistant tumor clone size were substantially larger than those from models in other cancer types predicting ~1 in 1 million cells to be resistant prior to treatment[9,32]. They were also consistent with our observation that a subset of the mutations comprising the resistant clone in P5 were present at a CCF of approximately 0.05 in the pre-treatment biopsy (Fig. 4b). The large size of the resistant clones prior to treatment suggested that they arose early in the evolutionary history of the tumor (Fig. 4c).

If resistant clones were present pre-treatment, the mutational processes[33,34] contributing subclonal mutations should be similar to those contributing mutations newly detected after treatment. Indeed, we found that while truncal mutations were associated with a unique set of mutational processes, subclonal and post-treatment mutations shared a similar set of mutational signatures (Supplementary Fig. 11). For example, in P5, the top two mutational signatures among truncal mutations, contributing to 71% of all such mutations, represented APOBEC activity, while the top two mutational signatures among both subclonal and post-treatment mutations, contributing to 66% and 72% of such mutations respectively, represented defective mismatch repair. The lack of a new signature post-treatment supported the view that the clones detected post-treatment were present pre-treatment, and suggested that the treatment itself was unlikely to be significantly mutagenic.

The tumors we analyzed post-treatment had, by definition, bulky residual disease and therefore likely larger resistant clone size prior to treatment than typical across the range of possible breast tumors. To more broadly characterize the evolutionary parameters that could lead to clonal replacement or other possible outcomes over a few months of neoadjuvant therapy, we used a stochastic mathematical branching model of exponential tumor growth[9,35–38] (Methods). Unlike our model of spatial tumor growth employed to assess between-region genetic divergence, this model was designed to assess the impact of baseline tumor sensitivity to therapy as well as rates of accumulation of resistance-causing aberrations on treatment outcome; these simulated tumors had no spatial structure. Each tumor was grown to 10 billion cells with a range of possible rates of accumulation of resistance-causing aberrations, and then treated for 150 days with resistant cells dying at the same rate as pre-treatment and sensitive cells dying at a range of possible rates. We defined four possible outcomes after treatment: pCR (<10,000 residual tumor cells), and if no pCR, sensitive residual disease (at least 80% of the residual tumor comprised of sensitive cells), clonal replacement (at least 80% of the residual tumor comprised of one resistant clone), and polyclonal resistant residual disease (less than 80% of the residual tumor comprised of sensitive cells, and multiple resistant clones) (Fig. 5a). Sensitive residual disease and pCR both occurred only at lower effective resistance aberration rates; sensitive residual disease occurred at the lower sensitive cell death rates and pCR at the higher sensitive

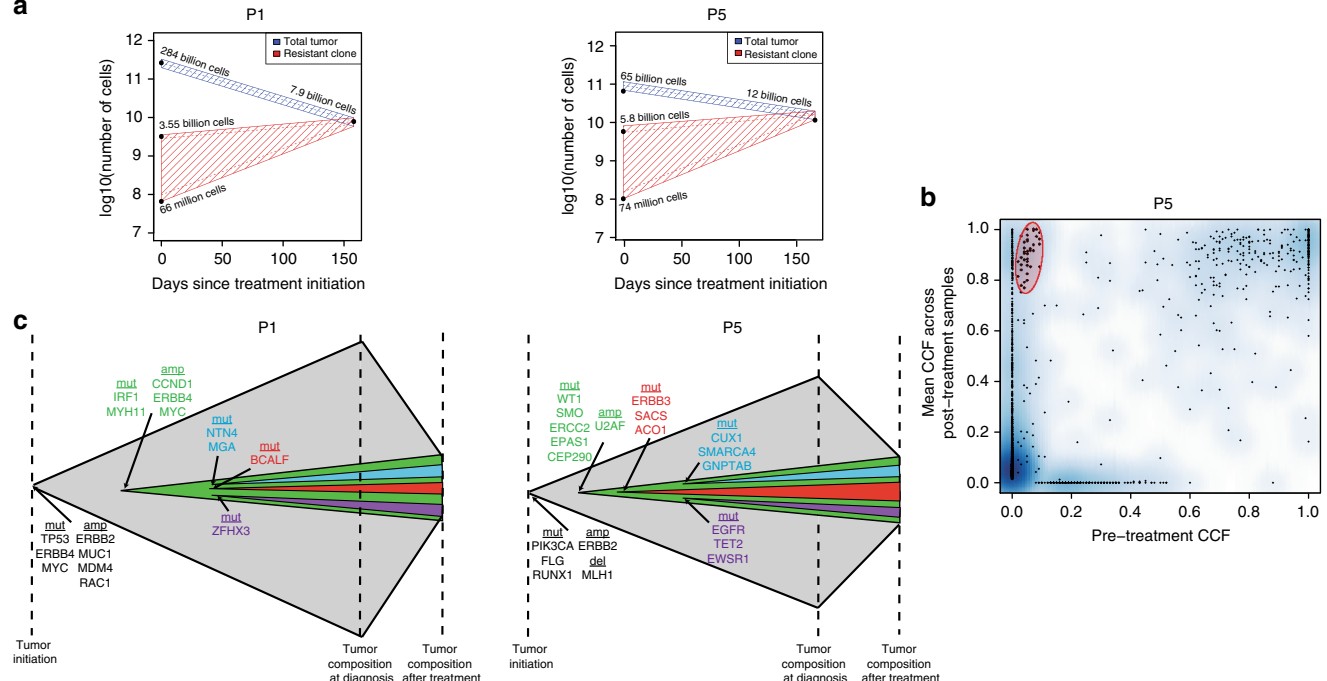

**Fig. 4** Growth of resistant subclones during therapy. **a** The change in overall tumor size (blue) and resistant subclone size (red) is shown for P1 and P5, assuming exponential growth and clonal replacement. Across a range of feasible growth rates, the resistant subclone was prevalent in the pre-treatment tumor. **b** In P5, a subset of mutations that were clonal in the post-treatment tumor sample were present at low frequency in the pre-treatment tumor sample (red oval). CCF cancer cell fraction. **c** Schematic for clonal replacement in P1 and P5, where width of the schematic over time corresponds to the logarithm of the number of tumor cells. Aberrations present in each subclone are listed: grey is truncal, green is the resistant subclone that replaces the tumor, and blue, red, and purple are subclones within the resistant subclone that become detectable post-treatment. Source data for panel **b** are provided as a Source Data file

cell death rates (Fig. 5b). At higher effective resistance aberration rates, polyclonal resistance and clonal replacement became increasingly frequent, with polyclonal resistance always more frequent than clonal replacement. Clonal replacement resulted when the time between the birth of the first and second resistant subclones was adequately long: this occurred at a stochastic rate of ~10% across a wide variety of effective resistance aberration rates (Fig. 5b).

We next used the mathematical model to infer the effective resistance aberration rates that would lead to the specific patterns observed in P1 and P5: clonal replacement with residual tumors 10–50% the pre-treatment size (Methods). For these tumors, we inferred effective resistance aberration rates between $6.0 \times 10^{-6}$ and $3.4 \times 10^{-4}$ (Fig. 5c). As a point of comparison, previous estimates of approximately 50 sites in the genome conferring resistance[9,35] and mutation rates of $10^{-9}$ to $10^{-7}$ [35,39,40] translate to effective resistance aberration rates of $5 \times 10^{-8}$ to $5 \times 10^{-6}$. Notably, with such a high effective resistance aberration rate, pCR would be impossible (Fig. 5b). Since both pCR and clonal replacement (with a large residual tumor) can occur, these simulation studies suggest that different breast tumors must possess markedly different effective resistance aberration rates, whether through higher rates of genomic change, higher numbers of sites in the genome conferring resistance, or both.

## Discussion

In this study, we assessed ITH of untreated and treated breast tumors, finding it to be on average higher and more widely variable in breast tumors than in other tumor types. The high ITH we observed in breast tumors could be related to high

selective pressure during primary tumor growth[17], as we model here, or potentially other factors such as differences in the microenvironment, spatial boundaries within the tumor, or differing modes of growth. Alternatively, technical differences in how the tumors were collected might have contributed to observed differences in ITH. Regardless of the cause, the presence of high pre-existing ITH meant that multi-region sampling of the post-treatment tumor was sometimes necessary to detect clonal replacement.

Our findings have implications in the era of personalized medicine as tumors increasingly undergo panel[41] or whole-exome sequencing[27] to identify potential targets of therapy or clonal antigens[42]. In particular, our results suggest that efforts to target therapies based on mutations in primary breast tumors, whether treated or untreated, would benefit from the analysis of at least two regions of the tumor. We find that, on average, one additional sample, even if from the same local region, adds the majority of clinically meaningful value over a single sample, though some additional value is gained with additional regions and with more geographically disparate sampling. In this study, we performed multi-region sampling of the treated tumors by sequencing different blocks from the surgical specimen, an approach that would be readily reproducible in future clinical studies.

We find that it is possible for the set of clonal mutations to shift dramatically over only a few months of combination chemotherapy and HER2-targeted therapy, such that one-third or more of the clonal mutations across the post-treatment tumor were rare in the pre-treatment biopsy. Such a shift occurred in two of the five tumors we analyzed, and we predict based on mathematical modeling that clonal replacement will occur in approximately 10% of cases across a variety of evolutionary

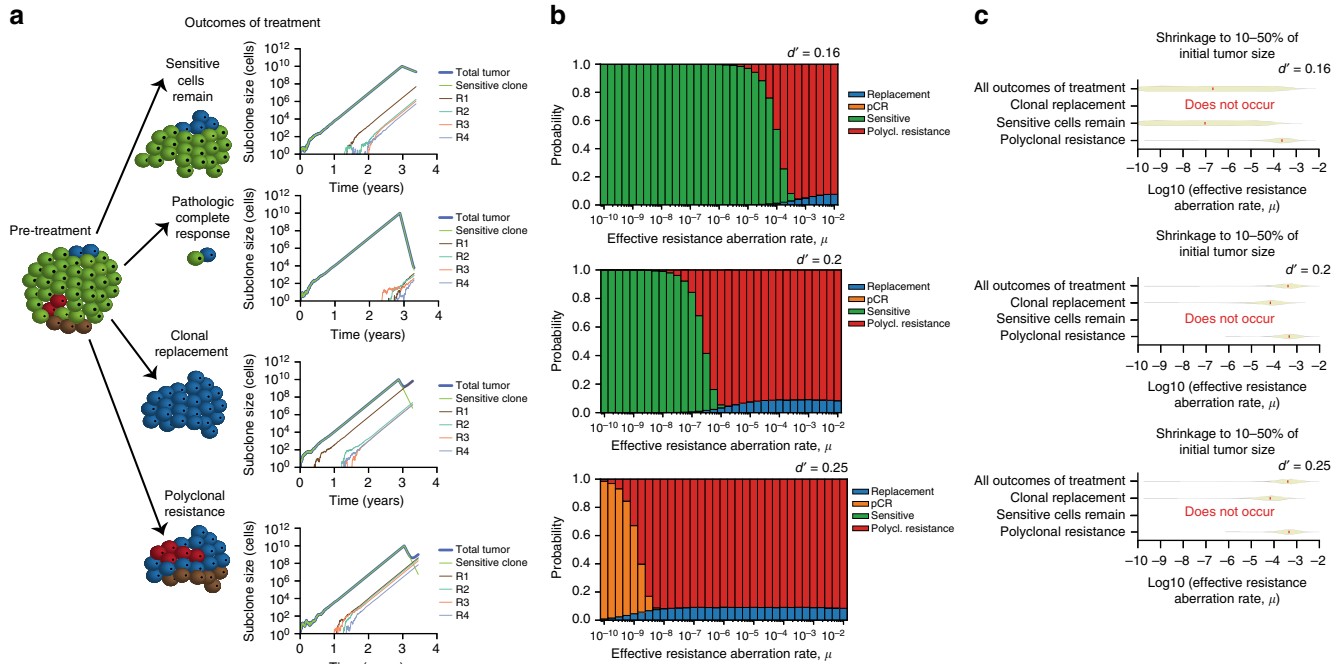

**Fig. 5** Evolutionary paths to different treatment outcomes. **a** Schematics of and example paths to the four defined treatment outcomes. In each simulation, the tumor grew to 10 billion cells and then was treated for 150 days. The size of the tumor and the first four resistant clones (R1–R4) are shown throughout primary tumor growth and treatment. **b** Stacked bar plots showing the proportion of simulated tumors with each of the four defined treatment outcomes for three plausible death rates of sensitive cells during treatment ($d' = 0.16$, 0.2, or 0.25) and varying effective resistance aberration rates ($\mu$, varied from $10^{-10}$ to $10^{-2}$ across the x-axis of each plot). The effective resistance aberration rate is the product of the rate of accumulation of genomic aberrations per site per cell division and the number of such aberrations that can confer resistance. Clonal replacement (blue) occurs at a rate of ~10% across many sets of parameters. **c** Inference of effective resistance aberration rates ($\mu$) for tumors that shrink to 10–50% of initial tumor size with treatment (matching the degree of tumor shrinkage observed in our cohort). We show inferred $\mu$ for the three plausible sensitive cell death rates ($d' = 0.16$, 0.2, or 0.25) and for possible treatment outcomes (clonal replacement, polyclonal resistance, and sensitive residual disease). For such large residual tumors to occur, either the sensitive cell death rate ($d'$) must be low, in which case clonal replacement cannot occur, or the effective resistance aberration rate ($\mu$) must be high. Source data for panels **b** and **c** are provided as a Source Data file

parameters. The frequency with which we observe and predict that such a drastic change occurs across neoadjuvant treatment suggests there may be some clinical benefit to characterizing the post-treatment tumor rather than the pre-treatment tumor in future clinical studies exploring adjuvant targeted therapy based on genomic biomarkers. Analogously, discordance between the primary tumor and the metastasis with regard to ER, PR, or HER2 status (in 10–50% of cases)[3,43] and actionable drivers (in >50% of cases)[44,45] has led to the recommendation to biopsy metastatic sites at recurrence to help determine appropriate treatment[46]. Furthermore, among those patients who have high-risk disease after neoadjuvant treatment, including those in this cohort, there may be prognostic implications to the different evolutionary paths their tumors have followed.

We inferred high effective resistance aberration rates in the two tumors that underwent clonal replacement, representing either high rates of genomic alteration (through mutation, copy number change, or epigenetic modification), or high numbers of potential resistance-causing aberrations, or both. In other words, it may be remarkably easy for certain tumors to acquire resistance in the context of their (epi)genomic and microenvironmental background. Other tumors, including those that undergo pCR, appear to be fundamentally different from these tumors from their earliest stages of development, both in how sensitive their founding cells are to the treatment, but also in their ability to acquire resistance. One explanation would be that tumors with different genomic backgrounds (such as the presence or absence of a certain driver mutation or copy number aberration) have

differing paths, and differing numbers of paths, to therapeutic resistance. We note that it is possible that factors not modeled here—including spatial differences in the impact of treatment or a subset of resistance-conferring changes that also confer a selective advantage in the absence of treatment—may also affect tumor outcome.

Approximately one-half of the tumors we initially profiled pre- and post-treatment were not analyzable with bulk whole-exome sequencing because of low cellularity, consistent with previous work in triple-negative breast cancer[8]. In other words, the evolutionary trajectories of the majority of HER2-positive breast tumors that either experience pCR or significant reductions in cellularity can be modeled, as described here, but not directly assessed at time of surgery. Earlier time points of sampling—for example, with tissue biopsies or cell-free DNA—may shed light on the patterns of evolution that lead to these favorable responses, as well as their impact on tumor heterogeneity. Similarly, using bulk multi-region whole-exome sequencing data, we were only able to identify with confidence tumors that had undergone clonal replacement: we were not able to distinguish between post-treatment tumors composed of multiple resistant clones (polyclonal resistance) and those comprised of mostly sensitive residual disease, though the mathematical modeling suggested that polyclonal resistance may be more common than clonal replacement. Delineation of polyclonal resistance would require multi-region sampling of the pre- as well as post-treatment tumor, ideally with concomitant single-cell profiling, to detect shifts in subclones across treatment.

In conclusion, multi-region sampling from untreated and treated primary breast tumors revealed substantial heterogeneity throughout treatment with chemotherapy and HER2-targeted therapy, even while major clonal sweeps took place in a minority of tumors. Many tumors that do undergo clonal sweeps have very large resistant clones present prior to treatment. The high inferred resistance aberration rates in these tumors suggest that additional studies of post-treatment tumors may uncover far greater numbers of mechanisms of therapeutic resistance than previously known. Multi-region sampling of breast tumors both pre- and post-neoadjuvant therapy may reveal the tumor's evolutionary path and, especially as increasing numbers of molecular and immune therapeutic targets are identified, provide clinical benefit.

## Methods

**Patient cohort, sequencing, and multi-region sampling**. The institutional review boards (IRBs) of the University of Southern California and Stanford University approved the collection and analyses described in this study. The study complied with all ethical regulations for work with human participants; informed consent was waived by the IRBs given analysis of archival, retrospectively collected samples. We collected formalin-fixed paraffin-embedded (FFPE) core diagnostic biopsies and surgical specimens from 21 patients with HER2-positive breast cancer who were treated at the University of Southern California: 20 of these patients received neoadjuvant therapy in between the time the core needle biopsy and surgical specimens were obtained without achieving a pCR, and one patient (P6) was not treated prior to surgery. We also obtained normal breast tissue from the surgical specimen or normal lymph node for each case. DNA was isolated using the QIAamp DNA FFPE Tissue Kit (Qiagen) and subjected to library preparation using the Agilent SureSelect Human All Exon kit (Agilent). From each region, 200 ng of DNA was used for whole-exome sequencing. Paired sequencing reads were aligned to human reference genome build hg19 with BWA 0.7.10.

We imposed stringent quality filters to ensure robust downstream analyses: in particular, reduction in cellularity with treatment can lead to bias when comparing pre-treatment and post-treatment tumor samples. We removed cases with, after read duplicate removal, <10× mean coverage in the normal sample or <15× mean coverage in the tumor ($n = 3$), cases where both pathology and computational (PurBayes[47]) estimates of purity were <50% ($n = 7$), and cases with <10 mutations in the post-treatment sample with a variant allele fraction (VAF) > 0.1 (ref. [48]) ($n = 2$) (Supplementary Fig. 5). An additional three tumors that passed these quality filters did not have additional regions from the surgical specimen for analysis and were not studied further.

For the untreated and five remaining treated HER2-positive tumors, we obtained an additional 1–5 regions from the surgical specimen for sequencing as above, with each region selected from a different, spatially distinct FFPE block of tumor tissue (Supplementary Data). All sections were reviewed by a board-certified pathologist (CJS) to select areas of high cellularity, with microdissection performed to increase estimated cellularity above 80%, wherever possible. These regions were sequenced as above, except that library preparation was performed with the Illumina Nextera Rapid Capture Exome kit (Illumina).

For analysis of untreated primary breast tumors, we selected 2–3 regions from each of three FFPE tumors from the METABRIC cohort[49] that had available matched normal tissue; one region was a core from a tumor block and 1–2 regions were spatially disparate sections from the same block (Supplementary Data). Sequencing was performed as above with the Agilent SureSelect Human All Exon kit. In selecting multi-region sampled whole-genome or whole-exome sequenced untreated breast primaries from the existing literature to analyze, we required estimated tumor purity of at least 40% and at least 40 mutations with coverage of at least 20 reads across all regions to ensure accurate assessment of mutational differences between regions (Supplementary Data).

**Variant and copy number calling**. Somatic single-nucleotide variants (SNVs) were called using Mutect 1.1.7 (ref. [50]). We then followed the variant assurance (filtering and rescuing) pipeline (VAP)[17]. In addition to stringent filtering for FFPE and other artifacts, the VAP uses multi-region sequencing data to borrow information across tumor regions to salvage false-negatives due to limits of the variant caller. VAFs were then calculated for the detected and rescued variants by dividing the number of reads carrying the variant by the total number of reads spanning that position. Mutations covered by less than 20 reads in any sample were removed, as were mutations where the alternate allele was not supported by at least four reads in at least one sample. Short insertions and deletions were detected using Strelka[51] (isSkipDepthFilters = 1). We salvaged filtered indels when the identical indel was identified (and not filtered) in other samples from the same case.

TitanCNA[52] was used to determine local copy number and purity of the tumor samples. For tumors P1–P6, the ploidy input was set based on the average value of

four centromeric FISH probes (chromosomes 8, 10, 11, and 17) measured on 100 cells in one sample from the surgical specimen. We assumed a single clone. Purity estimates from Titan (copy number-based) and Treeomics[53] (mutation-based) were compared as validation (correlation 0.92). Observed VAFs were adjusted for local copy number and purity following the framework presented in CHAT[54] to generate CCF estimates for each mutation in each sample. The nine new tumors in this study were analyzed in this way, as were three of the tumors from existing cohorts[14,16]. The non-breast (colon, brain, lung, and esophageal) tumors used as comparison in Fig. 1 were also processed using the same pipeline. For the other eight tumors from existing cohorts, we used the published CCF values[2,15]. Results for the nine tumors new to this study are available (Supplementary Data). Genomic data are deposited in the European Genotype Phenotype Archive (EGA): EGAD00001004306.

We used a robust workflow to identify copy number aberrations that were uniquely present before or after treatment in the tumors that underwent clonal replacement (P1 and P5). First, to define the post-treatment copy number aberrations, we used the highest quality post-treatment tumor sample, defined as the sample with the lowest average variance between log-ratio values within each Titan-called copy number segment. We did not use all the post-treatment regions due to differences in quality of the copy number calls derived from the whole-exome data and evidence for copy number similarity among the post-treatment regions in P1 and P5 as described in the Results (Supplementary Fig. 10). We defined a copy number segment as amplified in one time point and not the other if the segment was at least 100 base pairs in length with absolute copy number 4–8, with the amplified sample having at least twice the absolute copy number value of the other sample.

**Measurement of intra-tumor heterogeneity**. We measured pairwise mutational heterogeneity between samples using Fst[17,18] as well as HFR (Supplementary Fig. 1A).

Fst was computed per case by averaging the pairwise Fst statistics computed for each pair of samples in the case and using all subclonal SNVs (those with CCF < 0.5 in at least one sample, with CCF cut-off chosen because of its good performance in defining subclonality based on simulated virtual tumors[14]). SNVs with varying patterns of loss of heterozygosity between regions were not included in the pairwise Fst calculations[14]. For each pairwise comparison between regions, SNVs were defined as being either shared or region specific. For two regions $a$ and $b$, mutations $m_1$ to $m_t$,

$$\text{Fst}^{\text{Hudson}} = \frac{\sum_{m=1}^{m^t} (f_a^m - f_b^m)^2 \frac{f_a^m \times (1 - f_a^m)}{d_a^m - 1} - \frac{f_b^m \times (1 - f_b^m)}{d_b^m - 1}}{\sum_{m=1}^{m^t} f_a^m \times (1 - f_b^m) + f_b^m \times (1 - f_a^m)}, \quad (1)$$

where $f_a^m$ is the VAF for SNV $m$ and $d_a^m$ is the sequencing depth for SNV $m$ in region $a$[14].

HFR we define as follows:

$$[R1/(R1 + C)] + [R2/(R2 + C)]/2, \quad (2)$$

where R1 is the number of mutations with CCF > 0.5 in sample 1 and CCF < 0.1 in sample 2; R2 is the number of mutations with CCF > 0.5 in sample 2 and CCF < 0.1 in sample 1; and C is the number of mutations with CCF > 0.5 in both sample 1 and sample 2.

To measure heterogeneity between one sample and multiple other samples collected at a different timepoint, we define tHFR (temporal HFR) (Supplementary Fig. 1B) as follows:

$$tR/(tR + tC), \quad (3)$$

where tC is the number of mutations with CCF > 0.5 in all samples; and tR is the number of mutations with CCF > 0.5 in each of the second set of samples (here, post-treatment) and CCF < 0.1 in the first sample (here, pre-treatment).

We used the MEDICC[28] pairwise distance calculations to measure copy number heterogeneity between pairs of samples.

We used PyClone[55] to define mutational clusters and assess changes in cluster frequencies across samples. PyClone's Dirichlet process clustering was carried out on the filtered list of mutations for each sample. The pyclone beta binomial model was run using default parameters with 10,000 iterations with a burn-in of 1000. Due to a greater number of mutations, cases P1 and P5 were run for 20,000 iterations with a burn-in of 2000 and analysis was limited to the 1000 mutations with the highest coverage. For visualization of each case, we plotted PyClone clusters comprising at least 1% of the total number of utilized mutations.

**Classification of mutations and mutational signatures**. We classified mutations as driver variants (variants that confer a proliferative advantage leading to outgrowth of the subclone containing them) based on a previously curated list in breast cancer[26] or target variants (variants in genes that may have therapeutic or prognostic implications for patients) using the TARGET database[27]. We used Polyphen-2 (ref. [56]) to predict which missense mutations were damaging. For those mutations that were unique to and clonal within the post-treatment tumor in P1 and P5, which underwent clonal replacement, we used Cancer Genome Interpreter

to annotate plausible drivers of resistance[57]. We used the SomaticSignatures package[58] to identify COSMIC mutational signatures[33,34], dividing mutations into those that were truncal (mean CCF > 0.6 across all regions, and >0.25 in each region), subclonal (present in the pre-treatment sample with CCF < 0.4, or absent in at least one post-treatment sample), or post-only (absent in the pre-treatment sample). We analyzed the mutation sets from P1 and P5 separately, and grouped P2, P3, and P4 together given smaller mutation loads in these tumors.

**Spatial computational modeling**. We used a spatial agent-based model[17,20] to simulate tumor growth and mutation accumulation. We simulated $10^9$ cell virtual tumors with a range of selection coefficients ($s = 0$; 0.05; 0.1; 0.2; 0.4; 0.5) ranging from effectively neutral evolution to strong selection, and also a range of deme sizes (1000; 5000; 10,000; 50,000). The simulated tumors grew under a model of glandular fission; each freely mixing deme corresponded to a neoplastic gland that filled to its prespecified size and then divided into two demes that continued to grow. We recorded the mutational frequency distributions from a total of 20 regional samples taken across all octants of the tumor to allow for subsampling when calculating summary statistics; each sample was composed of ~$10^6$ cells. The deme-based spatial computational model used here is well established[17,20,59] and parallels the glandular structures often found in epithelial tumors[60]. In line with our previously described model[17,20], our simulated tumors increase in size in a peripherally dominated growth model, consistent with studies describing the higher degree of proliferation present in peripheral cancer cells compared to those at the tumor core[61]. During each cell division, somatic alterations arose via a Poisson process at a rate of $2 \times 10^{-8}$ per base pair, with a probability of conferring a driver (selectively advantageous) phenotype of $10^{-5}$. We used previously established values for the growth parameters (a birth rate of 0.55 and a death rate of 0.45, somatic alteration rate (0.6 new mutations per cell division across the exome), and driver alteration rate (1 in $10^5$ mutations causing a growth advantage)[17]. In line with the genomic data from our clinical cohort, for each tumor, we assumed a mean sequencing depth of 100×, and assigned 50 SNVs to represent pre-transformation truncal mutations. In calling mutations, we used the same criteria regarding number of total reads and variant reads as for the patient tumors.

We used ITH metrics as summary statistics for inference in conjunction with our simulated tumors: fHsub, fHSs, fHrs, Fst, and KSD (Code availability)[17]. We used Approximate Bayesian Computation (ABC)[20,62,63] to infer the deme size and selection coefficients for the primary breast tumors. Specifically, we obtained posterior parameter distributions for both deme size and selection coefficients by comparing the summary statistics derived from simulations encompassing the entire range of deme sizes and selection coefficients to those from the clinical cases with multi-region sampled genomic data within the ABC framework[64]. Given that this analysis indicated that neutral evolution and a deme size of 50,000 were unlikely to have led to the patterns of ITH observed in the primary breast tumors, in subsequent analyses we included only simulated tumors with $s = 0.05$; 0.1; 0.2; 0.4; or 0.5, and deme size = 1,000; 5,000; or 10,000.

**Mathematical modeling**. In estimating the size of the resistant clone prior to treatment, we assumed exponential growth with growth rates ranging from 0.65% to 2.8%. This corresponds to the median to upper 95% confidence interval growth rates and is based on two studies of HER2+ breast tumors[29,30]; we used higher growth rates given the large size of the tumors studied (we note that lower growth rates would lead to even larger estimations of the resistant clone size at baseline). We also took into account a ±25% uncertainty in volumetric tumor measurements. We assumed that the resistant clone represented 90–100% of the overall post-treatment tumor size, consistent with PyClone mutational clustering results (Supplementary Fig. 7). To convert between volume and cell-based tumor sizes, we used the approximation that a 1 $cm^3$ contains 1 billion cells[65,66].

We used a continuous-time, multi-type branching process to model tumor growth and evolution with treatment[9,35–38,67]. During the growth phase, all cells divide at rate $b = 0.15$ and die at rate $d = 0.13$, setting a growth rate of 2% per day, consistent with that observed in multiple studies of breast cancer[29–31]. Tumor growth starts from a single sensitive cancer cell. Cells accumulate genomic aberrations that confer resistance at a rate of $\mu$, and we consider a range from $10^{-2}$ to $10^{-10}$. The growth phase continues until the tumor reaches 10 billion cells. The tumor is then treated for 150 days, consistent with the duration of treatment observed in our neoadjuvant treated cohort. During treatment, no new mutations are generated; resistant cells continue to die at rate $d$, while sensitive cells die at a rate of $d'$ ($b < d'$), which we set to be 0.16, 0.2, or 0.25. For each set of parameters, we performed 100,000 simulations, and noted the size (in cells) of the total tumor and of each of the resistant clones throughout each simulated tumor's growth and treatment. To infer the most likely effective resistance aberration rate for P1 and P5, we assessed for each $d'$-value' the $\mu$-values that would lead to both clonal replacement (>80% of the residual cells deriving from one resistant clone) and shrinkage of the tumor to 10–50% of its initial size, and report the 5–95% interval of these $\mu$-values.

**Code availability**. Code used for mutation filtering, assessment of heterogeneity, and simulation studies is available on: https://github.com/cancersysbio/BreastCancerITH.

**Reporting summary**. Further information on experimental design is available in the Nature Research Reporting Summary linked to this article.

## Data availability

Whole-exome sequencing data generated for this study are deposited at the European Genotype Phenotype Archive (EGA) at EGAD00001004306. Data from previously published studies are available at: EGAS00001002153, EGAD00001000965, EGAD00001000898, EGAD00001001394, EGAD00001000714, EGAD00001000900, EGAD00001000984, EGAD00001001113, EGAS00001002947, and EGAS00001002737. The source data underlying Figs. 1, 3, 4b and 5b, c, as well as Supplementary Figs. 2–4, 6, 7, and 9–11, are provided as a Source Data file.

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

## Acknowledgements

We thank research advocates Susie Brain, Diane Heditsian, and Vivian Lee for helpful discussions. We gratefully acknowledge the TRACERx Consortium and Cancer Research Technology Limited for making genomic data available. This research was funded by the National Cancer Institute at the National Institutes of Health (R01CA182514 to C. Curtis) and Susan G. Komen (IIR13260750 to C. Curtis and PDR17481769 to J.L.C.-J.). C. Curtis is also supported by a Breast Cancer Research Foundation award. J.L.C.-J. is supported by a Damon Runyon Physician Scientist Training Award and a Stanford Cancer Institute Fellowship Award. J.G.R. is supported by an Erwin Schrödinger Fellowship (Austrian Science Fund FWF J-3996). Z.H. is supported by an Innovative Genomics Initiative Postdoctoral Fellowship. J.L. is supported in part by a National Institutes of Health grant P30CA014089. M.F.P. is supported by grants from the Breast Cancer Research Foundation and Tower Cancer Research Foundation. The acquisition of some of the samples was funded by the Department of Defense (W81XWH-10-BCRP-IDEA to D.T.).

## Author contributions

R.S. and J.G.R contributed equally to this work. J.L.C.-J., K.M., and C. Curtis designed the study. S.-F.C, H.B., C. Caldas, E.P., D.T., and J.L. selected and supplied the clinical specimens. J.L.C.-J., A.R., V.F., and J.L. performed clinical chart review. Z.M. processed the clinical specimens and generated sequencing data. J.L.C.-J., K.M., R.S., S.T., and J.D. processed the data. J.L.C.-J., K.M., and R.S. performed statistical analysis. K.M., J.G.R., and Z.H. performed simulation studies. R.W., E.P., C.J.S., and M.F.P. conducted pathologic review of the clinical specimens. J.L.C.-J., K.M., and C. Curtis interpreted the data. J.L.C.-J., K.M., and C. Curtis wrote the manuscript. All authors reviewed and provided feedback on the manuscript.

## Additional information

**Competing interests:** J.L. reports research funds from ANGLE Parsortix and is a member of the speaker bureau of Genomic Health. M.F.P. holds consulting or advisory role with honoraria at Karyopharm Therapeutics, Puma Biotechnology, Biocartis, Eli Lilly & Company, Novartis Pharmaceuticals, F. Hoffmann La-Roche Ltd. C. Caldas is a Scientific Advisor to Astrazeneca-iMed and has received research funding from

Astrazeneca, Servier, Genentech/Roche. M.F.P. has received research grants to his institution from: Cepheid, Eli Lilly & Company, Novartis Pharmaceuticals, F. Hoffmann La-Roche Ltd. C. Curtis is a Scientific Advisory Board member and shareholder of GRAIL and consultant for GRAIL and Genentech. The remaining authors declare no competing interests.

