## [Peer Review File · Nature Communications]

Reviewers' comments:

Reviewer #1 (Remarks to the Author):

This manuscript presents a study of the multiclonal nature of large breast tumours and compare it to simulations. This allows to discriminate to a certain extent between different scenarios of tumour progression and treatment effect. The manuscript is in general interesting and well written, although some points should be improved (see below).

Major points

1) I found the results in Fig 1AB showing that tumour heterogeneity is much higher for breast cancers as compared to other cancer types puzzling. Can the authors propose an explanation for this difference? Is that difference could be explained due to purely technical issues? It would be worth to have some elementary information regarding the non-breast tumours similar to what is found in Table S1 in order to rule out some explanations (e.g. much smaller tumours, very different number of mutations, different number of regions analysed etc).

2) The distribution of tHFR is very dependent on the number of regions sequenced (Fig 3D). For instance, a patient with 3 regions sequenced should be compared with the distribution with 3 regions also. So, the patient-specific number should be indicated on the figure, and were the simulations (lines 249-250) made taking that into account?

To further drive the point that one sample is not enough, but two samples are (lines 254-255), the actual distribution of tHFR in function of the number of regions sequenced per patient should be shown (i.e. what happens if only one of the samples is kept for P1, if two samples are kept, etc). With this, if the point that 2 regions are necessary and sufficient is correct, then P1 should not be flagged as exceptional with one sample but should be with any two samples.

3) Line 302 - " ~ 1 in 1 million cells to be resistant prior to treatment". Since the samples were chosen because they had a large enough residual disease to allow exome sequencing, they may be quite different from a "typical" tumour, and a much larger fraction of resistant cells may be expected.

4) If I'm not mistaken, the mathematical modelling used for the simulation of resistance is completely different from the spatial computational modelling used in the rest of the manuscript.

Why? Also, assumptions like "all cells divide at rate $b = 0.15$ and die at rate $d = 0.13$ " imply a constant growth rate in the whole tumour, which is at odds with the spatial modelling approach.

5) 341-352 - the conclusion is that "different breast tumours must possess markedly different effective resistance aberration rates" - interesting, but at the same time it means that the method does not really accurately predict what is observed. Why? Proposed explanations: higher rates of genomic change  unlikely, as they have about the same number of mutations. Higher numbers of sites in the genome conferring resistance  where would that difference come from? If the difference is not due to mutations, then what could it be? And if it is due to mutations, then it's a circular argument of kind. Tumour microenvironment - maybe, but what really? And isn't this incompatible with the (poly-) clonal selection observed?

Some ideas - more than one mutation is necessary (in some cases)? Mutations offering resistance also give small fitness boost?

Minor points

6) It is not readily clear that the tumours in Table S1 annotated as "treated" are only used in the second part of the manuscript, while the tumours annotated as "untreated" are only used in the first part of the manuscript. This is particularly disorienting because the treated tumours are the first in the table. It would be better to have an extra column stating in which part of the analysis each tumour was used. Finally, for those treated, there should be coverage, purity values and mutation count for before / after treatment - I think this is partially the case, but it's not clear what is what.

7) F_{st} should be defined in the text.

8) A deme is defined as a well-mixed tumour cell subpopulation. That's quite unclear to me, and also, it is unclear whether this is a purely technical parameter or if it has any biological significance. Deme size is relevant to the quality of the fit (Fig 2S B), but why is it important? Does it mean anything?

9) Bayesian posterior probability calculation should be detailed. Bayesian posterior probability should not be used to discriminate hypotheses in general, although it may be OK in the setting where it is used (line 159).

10) Figure S4 - does it make sense to make the calculations for all selection coefficients at the same time? Are the results more or less independent of the selection coefficients?

11) There are only 8 tumours in Fig 1a compared to 9 in Fig 1b. I suppose one is hidden below another. A different representation avoiding the issue would be better.

12) Line 493 - there's an extra "4" at the end of the line.

Reviewer #2 (Remarks to the Author):

Dear editor,

“Clonal replacement and heterogeneity in breast tumors treated with neoadjuvant HER2-targeted therapy”

Caswell-Jin et al. describe in this manuscript the assessment of intra-tumor heterogeneity (ITH) by analysing whole exome sequencing (WES) data from treated and non treated multi-region formalin fixed paraffin embedded breast cancer samples to reveal the evolutionary tract of resistance mechanism of HER2 targeted treatment.

Overall the manuscript is well written and performed by strong multidisciplinary international research groups. The authors provide a structure to assess ITH in Formalin fixed and paraffin embedded tissues which could be of great potential use in daily clinical practice of pathology departments. Their main finding is that, although the number of analysed tumors (n=14) is low in this study, the authors showed that efforts to target therapies based on mutations in primary breast tumors would benefit from analysing more than one sample per tumor. They showed that on average approximately 30% of the clonal mutations observed in one biopsy would be absent in the other simply due to the ITH present prior to treatment.

In general, the authors provide additional data that clonal mutations can change significantly over a short period of time after neoadjuvant chemotherapy, though my major concern is on the low number of analysed PRIMARY TUMORS and only one untreated HER2 positive tumor to draw conclusions. nevertheless this manuscript will arise awareness about ITH in breast cancer that should be studied more to understand resistance mechanism in HER2 positive breast cancer. This

information is of high clinical potential and may have large implications in the era of personalized medicine. Therefore I would consider this work for publication, nevertheless, there are major and minor concerns which should be addressed.

Concerns

Major: 1. Line 142: Please comment on whether analyzing only 1 untreated HER2 positive tumor for which multi-region samples from primary breast tumors was available would be enough to understand the potential impact of ITH in HER2 targeted treatment induced genomic changes, and to draw the conclusions. A few additional cases of untreated Her2 positive primary breast cancer would add a lot of strength to the conclusions.

Major: 3. Line 212: HFR was similar between multi-region sampled pre- (30%, range 1-70%) and post- 213 (28%, range 10-54%) treatment tumors “: Please comment whether this is perhaps expected as the analyzed tumors where almost all pT3 tumors and did not respond to neoadjuvant chemotherapy, so similar HRF is expected?

Major: Line 262-264” and observed that the degree of clonal change seen in P5 was statistically unlikely to have resulted from pre-existing heterogeneity (tHFR > 99th percentile). In the previous section Line 249-250 the argument was made that taking only 1 biopsy ; 30% (P5: tHFR = 0.3 > 90th percentile) of the clonal mutations would be absent in the other biopsy. Maybe I don’t understand, but how do these analyses relate to each other? PLease comment.

Major: Line 73-76 “Such an evolutionary model would have implications for treatment, including whether adjuvant therapies should be targeted based on the genetic composition of the post-treatment rather than the pre-treatment tumor.” And Line 382 and 384: “The frequency with which ... adjuvant therapy”

I guess that the authors are implying here the setting of adjuvant treatment which if given in patients treated with neo-adjuvant treatment and surgery? By inquiring our medical oncologists adjuvant treatment decisions in breast cancer are already made based on pre- and post treatment tumor characteristics. This treatment options and decisions are made during multi-disciplinary meetings where surgeons, medical oncologist, pathologists and radiologists discuss breast cancer patients. Please comment.

Minor: 2. Line 202: It is difficult to follow how many of the 20 primary Her2 positive breast biopsies (pre-treatment) had interpretable bulk whole-exome sequencing. Please comment / add to the manuscript or add to figure suppl 5 whether the drop out of 45%, was only due to low cellularity at time of surgery or due to low quality DNA of the pre-treatment biopsy or low cellularity of the pre-treatment biopsy.

Minor: Line / Paragraph 268-277: This might be a difficult paragraph to follow for the reader. What is the added value of this paragraph if there is only a trend and not significant t-test between the pre- and post treatment samples of the tumors that have undergone clonal replacement compared to tumors that didn't undergo a clonal replacement. Please comment. Might help to add MEDICC53 ref in this paragraph.

Minor: LINE / Paragraph 279-292: The analyses in this paragraph are completely underpowered to make any conclusions about potential causes of resistance to HER2-targeted therapy combined with chemotherapy. Please comment and how do these results relate to earlier published results?

Response to Reviewers

We are grateful to the reviewers for their thorough assessment of our work and many insightful comments. We are also pleased to see that they found our work to be “interesting and well written”, and “of high clinical potential” with possibly “large implications in the era of personalized medicine”. Indeed, our hope is that in outlining the polyclonal nature of large breast tumors across neoadjuvant therapy, we may establish best-practice, feasible approaches to sampling breast tumors, measuring their heterogeneity, and determining clonal mutations. It is clear, based on our work, that distinct regions of a primary breast tumor may exhibit vast differences in apparently clonal mutations, and that, in a subset of tumors, these clonal differences may shift further across therapy. We believe that an awareness of this regional and temporal variation will be essential in the success of future precision oncology efforts.

We agree with Reviewer #2 that a limitation of our work is the relatively small sample size. Two additional multi-region breast cohorts have now been published (Ullah et al, *J Clin Invest*, 2018, PMID 29480816 and Barry et al, *Clin Cancer Res*, 2018, PMID 29891724), and we have mined these such that our multi-region sequenced untreated tumors cohort is now N=15 rather than N=9. We agree with Reviewer #1 that it is difficult to be certain whether technical factors may have contributed to variability in intra-tumor heterogeneity, both within breast tumors and between breast and other tumors. We have explored these factors to the best of our ability, but hope to also motivate with our work more careful reporting of precise tumor collection and analysis strategies. Our responses to these and the other comments, which we believe have substantially improved our manuscript, are detailed below. To facilitate the review of these changes in the manuscript, we have denoted them with blue text.

Reviewer #1 (Remarks to the Author):

This manuscript presents a study of the multiclonal nature of large breast tumours and compare it to simulations. This allows to discriminate to a certain extent between different scenarios of tumour progression and treatment effect. The manuscript is in general interesting and well written, although some points should be improved (see below).

Major points

1) I found the results in Fig 1AB showing that tumour heterogeneity is much higher for breast cancers as compared to other cancer types puzzling. Can the authors propose an explanation for this difference? Is that difference could be explained due to purely technical issues? It would be worth to have some elementary information regarding the non-breast tumours similar to what is found in Table S1 in order to rule out some explanations (e.g. much smaller tumours, very different number of mutations, different number of regions analysed etc).

We agree with the reviewer that the explanation for the higher heterogeneity we observed among breast tumors relative to the other tumor types examined is not immediately obvious. In our original manuscript, we explored (through computational modeling) the possibility that differences in the parameters of tumor growth (especially the extent of subclonal selection) could explain these differences. Indeed, our models suggest that such a difference could explain the results. However, the reviewer raises the important point that technical differences in how the tumor regions were collected and analyzed across these diverse cohorts may have played a role. We have added details (where available) about these technical aspects for all tumors to Supplementary Table 1 (untreated breast tumors are in 1A, treated breast tumors in 1B, and the other tumor types are in 1C). We have also added the *HFR* measure of heterogeneity to this table to allow readers to directly interpret the associations between various technical factors and heterogeneity.

We did observe that the breast tumors had generally lower purity and coverage than the other tumor types, despite our stringent quality control criteria (mean purity 62.4% vs 77.7%, $P=2.6E-4$; mean coverage per mutation used 90X vs 115X, $P=0.05$). We did not observe a significant correlation between purity or coverage and *HFR* within breast tumors or within the other tumor types, but may have been limited by numbers: we note that amongst breast tumors, there was a non-significant trend toward lower purity tumors having higher *HFR* ($r^2=0.08$, $P=0.22$), and toward lower coverage tumors having higher *HFR* ($r^2=0.16$, $P=0.08$) (no correlation for either amongst the other tumors, given minimal variation in *HFR* between these tumors). In a multivariate model including type (breast vs other), coverage, and purity, only type remained significant ($P=0.01$). In other words, purity and coverage differences may abrogate, but do not entirely explain, the difference we observed.

The other variables we examined (number of regions examined, tumor size, whole-genome vs whole-exome sequencing, number of mutations per region [overall, or stratifying by whole-genome vs whole-exome], and fresh-frozen vs FFPE) did not correlate significantly with *HFR*, though one interesting trend (tumor size) is shown below. Counterintuitively, there appears to be a borderline negative correlation ($r^2=0.20$, $P=0.05$) between tumor size and *HFR* amongst breast tumors, with larger breast tumors unexpectedly, and quite likely by chance, trending toward lower *HFR*.

It is certainly plausible that other, unmeasured differences could exist between the cohorts. In reviewing the papers to find differences between the cohorts, we noted that relatively few reported the amount of DNA input for sequencing, which could have an impact on measured heterogeneity. We used 200 ng of input and had a mean *HFR* of 28% in our cohort; another study of breast tumors (Barry et al, 2018) used 500 ng of input and had a mean *HFR* of 20% (difference nonsignificant with small numbers, $P=0.26$). It is possible that sequencing smaller amounts of DNA per region could drive heterogeneity up, but we cannot say this with the data available to us. As we consider all the factors that could theoretically influence tumor

heterogeneity, it becomes increasingly apparent that precise reports of exactly how tumors are collected and analyzed will be essential as this is further studied.

We have added the information about purity and coverage differences between breast and the other cohorts to the Results section (page 4). We have also added to the Discussion section (page 9) acknowledgment that technical differences in how the tumors were collected may have contributed to observed differences in ITH.

2) The distribution of *tHFR* is very dependent on the number of regions sequenced (Fig 3D). For instance, a patient with 3 regions sequenced should be compared with the distribution with 3 regions also. So, the patient-specific number should be indicated on the figure, and were the simulations (lines 249-250) made taking that into account? To further drive the point that one sample is not enough, but two samples are (lines 254-255), the actual distribution of *tHFR* in function of the number of regions sequenced per patient should be shown (i.e. what happens if only one of the samples is kept for P1, if two samples are kept, etc). With this, if the point that 2 regions are necessary and sufficient is correct, then P1 should not be flagged as exceptional with one sample but should be with any two samples.

This is true. In general, as additional post-treatment regions are added, *tHFR* should decrease, and this decrease should be most dramatic going from one post-treatment region to two post-treatment regions in tumors that did not undergo clonal replacement (similar to how we find, in untreated tumors, that the proportion of apparently clonal mutations reclassified as definitively subclonal increases the most dramatically with a second profiled region, as illustrated in Figure 1D). For Figure 3D, we have done as the reviewer suggested, downsampling to a specified number of post-treatment regions and taking the average across all combinations, and reporting the appropriate value for each modeled distribution. The updated Figure 3D and related Supplementary Figure 9 are reproduced here:

Figure 3D

Supplementary Figure 9

As expected, all cases experience a reduction in *tHFR* as additional samples are added. Among patient tumors, P1 has the largest change across treatment even with one sample (only 1.5% of values in the simulated distribution higher than it); in other words, even with only one region post-treatment, P1 did have suggestion of clonal change. The strength of that suggestion, of course, increases with additional regions (0.3% of values in the “two region” specific distribution with two regions of P1, and 0.06% of values in the “three region” specific distribution with three regions of P1), even as the *tHFR* value itself decreases. In other words, in some cases (like P1),

the degree of clonal change can be so dramatic that it is detectable with one region. We have added commentary on this occurrence to the Results (page 7).

P3 (no clonal replacement) and P5 (clonal replacement) illustrate the need for multi-region sampling post-treatment. They have similar values of *thFR* with one region. With two regions, P3's *thFR* drops dramatically, ruling out clonal replacement, while P5's remains similar. At three regions, there is suggestion that P5 may have undergone clonal replacement (6% of values in the simulated distribution higher than it) that is confirmed in the sampling scheme-specific simulation (0.4% of values in the scheme-specific distribution higher than it). We have clarified that the value of multi-region sampling post-treatment is seen in P3 and P5 in the Results (page 7) and in legend of Figure 3.

3) Line 302 - "~1 in 1 million cells to be resistant prior to treatment". Since the samples were chosen because they had a large enough residual disease to allow exome sequencing, they may be quite different from a "typical" tumour, and a much larger fraction of resistant cells may be expected.

We agree it is almost certainly true that those tumors with adequately bulky residual disease to analyze with whole-exome sequencing after treatment possessed fundamentally different landscapes of resistance prior to treatment than those tumors we could not analyze post-treatment (because their residual disease had poor cellularity, or because they had no residual disease at all). Our intent in the mathematical modeling section that ends the Results was to turn from this calculation of 0.02%-12.5% resistant cells in the tumors we analyzed post-treatment to a more universal assessment of the range of possibilities. There, we indeed confirmed that the resistance aberration rates that would generate tumors that went on to achieve pathologic complete response were substantially lower from those in the tumors we analyzed. In other words, as the reviewer notes, across all tumors, a rate of 1 in 1 million resistant cells prior to treatment may in fact be a common (perhaps even average) occurrence. What our analysis of post-treatment tumors with bulky residual disease establishes is that it is not the *only* occurrence; that is, that there is a wide range of possible landscapes of resistance prior to treatment, which, in our models, is dictated by the rates of accumulation of resistance aberrations as determined early in tumor development. We have added a sentence to the Results (page 8) clarifying these points, which we believe also help motivate our use of the mathematical modeling approach, as discussed further in response to point #4.

4) If I'm not mistaken, the mathematical modelling used for the simulation of resistance is completely different from the spatial computational modelling used in the rest of the manuscript. Why? Also, assumptions like "all cells divide at rate $b = 0.15$ and die at rate $d = 0.13$ " imply a constant growth rate in the whole tumour, which is at odds with the spatial modelling approach.

Yes, this is correct, and we agree that our transition from one type of modeling to another, and our justification for it, should be clarified. As the reviewer notes, the two models have different assumptions: the mathematical model we use to simulate resistance throughout treatment assumes a freely mixing population with constant growth rate throughout the tumor, while the spatial model we use to simulate primary tumor growth does not allow free mixing and has varying growth rates throughout the tumor. Our rationale for these differences is that each model was designed to answer a different question, and we opted for the simplest model needed to address the question of interest.

(1) Spatial model:

- Time period simulated: primary tumor growth only
- Varying inputs: glandular structure (deme size) and selection coefficient

- Outcome of interest: **between-region genetic divergence** (absolute measures, and also whether a scenario is plausible where a small portion of the tumor is relatively homogenous within itself but highly divergent with another part of the tumor, which would have implications for analyses across treatment)

(2) Mathematical model:

- Time period simulated: primary tumor growth and treatment
- Varying inputs: rate of accumulation of resistance-causing aberrations and founding tumor cell sensitivity to treatment
- Outcome of interest: **treatment outcome** (pathologic complete response, clonal replacement, polyclonal resistance, sensitive residual disease)

In other words, while the spatial model generates spatial structure to assess between-region genetic divergence, a simpler model that does not include spatial structure can assess the impact of rates of resistance-causing changes or founder tumor cell sensitivity on treatment outcome. Would incorporating spatial structure change our results from those obtained with the simpler mathematical model? More explicitly, would clonal replacement occur with differing frequency and/or with differing rates of accumulation of resistance-causing aberrations than in the simpler mathematical model? We used our spatial model to evaluate the most direct way that spatial structure could change the conclusions drawn from freely mixing model – that is, if geographic differences in the impact of treatment on a spatially heterogeneous tumor could lead to apparent clonal replacement – and found such a scenario to be an unlikely explanation for the observed clonal replacement in our cohort. It is challenging to assess the possible impact of spatial differences on our other conclusions given that it is not immediately apparent – and the genomic data available in this cohort do not give us adequate granularity to assess – how treatment might affect different aspects of the tumor differently (for example, peripheral tumor cells dying more quickly than internal tumor cells, or tumor cells more proximal to modeled vasculature dying more quickly than those more distal). We believe the impact of spatial structure on treatment outcome is an important area of research, but felt that the simple mathematical model employed in this manuscript was adequate to support our conclusion: that the rate of accumulation of resistance-causing changes likely vary substantially between breast tumors, and must be quite high in some. We believe this result is robust to incorporation of spatial structure into the models.

We have added explicit clarification about the differences in the assumptions of the models to the Results (page 8), and commentary on this limitation to the Discussion section (page 10). More generally, we note that the complexity of possible scenarios and lack of ground truth data on human tumor evolution motivates the need for computational models of these processes towards the goal of informing improved study designs and future experimental efforts.

5) 341-352 - the conclusion is that "different breast tumours must possess markedly different effective resistance aberration rates" - interesting, but at the same time it means that the method does not really accurately predict what is observed. Why? Proposed explanations: higher rates of genomic change  unlikely, as they have about the same number of mutations. Higher numbers of sites in the genome conferring resistance  where would that difference come from? If the difference is not due to mutations, then what could it be? And if it is due to mutations, then it's a circular argument of kind. Tumour microenvironment - maybe, but what really? And isn't this incompatible with the (poly-) clonal selection observed? Some ideas - more than one mutation is necessary (in some cases)? Mutations offering resistance also give small fitness boost?

We do suspect that one or both of these explanations (higher rates of genomic change/genomic instability, and/or higher number of sites in the genome conferring resistance) account, at least in part, for the inferred markedly different effective resistance aberration rates between tumors.

We note that P1 and P5 did have the highest average mutation counts per region relative to the other tumors across treatment from our cohort as well as to the untreated whole-exome sequenced tumors we analyzed. In other words, greater genomic instability might explain, at least in part, the large size of the pre-treatment resistant clones in P1 and P5 relative to the other tumors. We do not however believe that variation in mutation rate (or in genomic instability as a whole) could be the sole factor explaining variation in rates of resistance; we note, for example, that one study of triple-negative breast cancer found that higher mutational burden correlated with a *higher* probability of pathologic complete response (Jiang et al, *PLoS Medicine*, 2016, PMID 27959926). We therefore suspect that the other component of the effective resistance aberration rate – the number of sites that confer resistance – may differ between tumors as well. The hypothesis here would be that tumors with certain driver alterations (e.g. a *PIK3CA* mutation or a *CCND1* amplification) would have a different number of paths to therapeutic resistance available to them than tumors with other driver alterations (e.g. a *TP53* mutation or a *MYC* amplification). This idea fits with the proposal by the reviewer that in some cases, more than one mutation may be necessary to confer resistance. That is, in our model, there could be one mutation present in the tumor's founding cell population, and because of the presence of that mutation, a second mutation would confer resistance that would not in the absence of that first mutation. This hypothesis would explain our findings in the variability in rates of resistance-causing aberrations without invoking rates of mutation or copy number change. We have added an outline of this hypothesis to the Discussion (page 10).

It is possible that two mutations (with the first not present in the founding tumor cell) could be necessary to confer resistance, a scenario that we do not model. Effectively, the rate of such a "resistance-conferring change" would be smaller than that of a one-hit change. It is likely that some "resistance-conferring changes" (whether epigenetic, two-hit mutations, or copy number alterations) are easier to achieve than others; we did not model variability in this rate across the range of possible resistance-conferring changes, but we do not think the existence of this variability would substantially alter our findings of variability in the average rate of these changes between tumors. The idea that mutations that confer resistance might also, in some cases, give a small fitness boost even prior to treatment is an intriguing one. Generally, the reverse is suspected to be true (e.g. Thomas et al, *PLoS Biology*, 2018, PMID 30278037): that is, the standard model is that resistance to therapy reduces fitness in the absence of therapy, while sensitivity to therapy increases fitness (for example, highly fit rapidly dividing tumor cells also die more rapidly with chemotherapy). For this reason, we did not incorporate a scenario where a resistance-conferring change conferred selective advantage prior to treatment into our models, but it is plausible that this may occur in some cases, and have added a comment on this possibility to the Discussion (page 10).

Minor points

6) It is not readily clear that the tumours in Table S1 annotated as "treated" are only used in the second part of the manuscript, while the tumours annotated as "untreated" are only used in the first part of the manuscript. This is particularly disorienting because the treated tumours are the first in the table. It would be better to have an extra column stating in which part of the analysis each tumour was used. Finally, for those treated, there should be coverage, purity values and mutation count for before / after treatment - I think this is partially the case, but it's not clear what is what.

We have restructured Table S1 such that S1a is now untreated tumors and S1b is treated tumors. We have also added a phrase in the Introduction (page 3) clarifying that the untreated and treated cohorts are separate and non-overlapping (a necessity because of the lack of multi-region pre-treatment samples available retrospectively on tumors that were treated neoadjuvantly). Within Table S1, we have clarified pre- vs post-treatment coverage and purity and have added the average mutation count per region pre-treatment and post-treatment as columns as well.

7) F_{st} should be defined in the text.

We have added the equation used for F_{st} to the Methods (page 13-14).

8) A deme is defined as a well-mixed tumour cell subpopulation. That's quite unclear to me, and also, it is unclear whether this is a purely technical parameter or if it has any biological significance. Deme size is relevant to the quality of the fit (Fig 2S B), but why is it important? Does it mean anything?

We thank the reviewer for highlighting this point of confusion. The spatial modeling framework that we extend in this manuscript and developed previously (in particular in Sottoriva et al *Nature Genetics* 2015 and Sun et al *Nature Genetics* 2017) operates under a model of glandular growth and fission where the 2017 paper extends this to the more general case of "deme" subpopulations. The biological equivalent of a "deme" here would be a neoplastic gland within an adenocarcinoma. Our simulated tumors grow and freely mix within these neoplastic gland units (e.g. of size 1,000 or 5,000 cells); once one unit is filled, it divides in two (i.e., glandular fission) and then continues to grow to fill each of the two new units, etc. Changing the size of these freely mixing glandular units of growth effectively changes the spatial constraint on the growing tumor: a smaller deme size allows less unconstrained growth, and therefore tends toward higher between-region genetic divergence, as illustrated in Figure S3. Thus we find that our breast tumors, with relatively higher between-region genetic divergence as measured by HFR , had inferred deme sizes typically less than 50,000 cells. We note that other modes of tumor growth conferring different spatial constraints are also plausible, though we chose this mode as we felt it was best supported by existing literature. For example, if a colon tumor grew by glandular fission (as modeled here) but a breast tumor grew by "spillover" (e.g. a "deme" filled with cells and then only a few, rather than half, moved into the next "deme"), such a scenario could similarly lead to higher HFR in breast tumors relative to colon tumors. We have added a more thorough description of the growth model (including the biological significance of a "deme") to the Methods (page 14), a phrase explaining that demes correspond biologically to neoplastic glands to the Results (page 5), and an acknowledgment that differing modes of growth could also change HFR between tumor types to the Discussion (page 9).

9) Bayesian posterior probability calculation should be detailed. Bayesian posterior probability should not be used to discriminate hypotheses in general, although it may be OK in the setting where it is used (line 159).

We have added a more detailed description of our posterior probability calculations to the Methods section (page 15), as well as references from previous publications from our lab and others that use the same inference framework (in particular, Figure S7a from our paper Sottoriva et al *Nature Genetics* 2015, describes our approach). We have provided the R code used to generate the posterior probabilities shown in Figure S2 in github as well.

10) Figure S4 - does it make sense to make the calculations for all selection coefficients at the same time? Are the results more or less independent of the selection coefficients?

That is correct: between-region genetic divergence (whether measured by HFR or F_{st}) is more or less independent of the selection coefficient in our models over multiple levels of "strong" selection ($s=0.05...0.5$). This is in contrast to neutral evolution ($s=0$), which generates markedly reduced patterns of between-region genetic divergence. This observation is explored in more detail in our paper Sun et al, 2017, *Nature Genetics*, PMID 28581503.

As regards Figure S4, key values are shown below by selection coefficient:

	$s=0$		$s=0.05$		$s=0.2$		$s=0.5$	
	Mean	sd	Mean	sd	Mean	sd	Mean	sd
2	0.029	0.073	0.347	0.251	0.321	0.230	0.270	0.226
5	0.045	0.078	0.418	0.229	0.411	0.211	0.358	0.210
10	0.061	0.080	0.451	0.217	0.454	0.192	0.400	0.190
15	0.069	0.081	0.468	0.210	0.472	0.180	0.421	0.174
20	0.076	0.081	0.476	0.205	0.480	0.175	0.432	0.166

There is a trend toward lower between-region genetic divergence at $s=0.5$ relative to $s=0.05$ or $s=0.2$ (this can also be seen in Figure 1D), but the magnitude of this difference is quite small, especially compared to the difference between between-region genetic divergence at any selection coefficient ≥ 0.05 and neutral evolution, also shown above. The situation is circular: we cannot distinguish whether a simulated model of tumor growth at various selection coefficients between 0.05 and 0.5 best fits the between-region genetic divergence seen in our patient tumors (as illustrated in Figure S2A) because between-region genetic divergence at these varying levels of non-neutral selection is similar. We do show the subtle differences by selection coefficient in Figure 1D, but for purposes of Figure S4, which we include to emphasize the large variability from tumor to tumor (which is true no matter what selection coefficient, as illustrated by the standard deviations above), we combined all non-neutral simulated scenarios. We have added a note to this effect in the caption of Figure S4.

11) There are only 8 tumours in Fig 1a compared to 9 in Fig 1b. I suppose one is hidden below another. A different representation avoiding the issue would be better.

We made the points smaller and added some jitter so that all points can be seen.

12) Line 493 - there's an extra "4" at the end of the line.

Thanks! fixed.

Reviewer #2 (Remarks to the Author):

Dear editor,

“Clonal replacement and heterogeneity in breast tumors treated with neoadjuvant HER2-targeted therapy”

Caswell-Jin et al. describe in this manuscript the assessment of intra-tumor heterogeneity (ITH) by analysing whole exome sequencing (WES) data from treated and non treated multi-region formalin fixed paraffin embedded breast cancer samples to reveal the evolutionary tract of resistance mechanism of HER2 targeted treatment.

Overall the manuscript is well written and performed by strong multidisciplinary international research groups. The authors provide a structure to assess ITH in Formalin fixed and paraffin embedded tissues which could be of great potential use in daily clinical practice of pathology departments. Their main finding is that, although the number of analysed tumors (n=14) is low in this study, the authors showed that efforts to target therapies based on mutations in primary breast tumors would benefit from analysing more than one sample per tumor. They showed that on average approximately 30% of the clonal mutations observed in one biopsy would be absent in the other simply due to the ITH present prior to treatment.

In general, the authors provide additional data that clonal mutations can change significantly over a short period of time after neoadjuvant chemotherapy, though my major concern is on the low number of analysed PRIMARY TUMORS and only one untreated HER2 positive tumor to draw conclusions. nevertheless this manuscript will arise awareness about ITH in breast cancer that should be studied more to understand resistance mechanism in HER2 positive breast cancer. This information is of high clinical potential and may have large implications in the era of personalized medicine. Therefore I would consider this work for publication, nevertheless, there are major and minor concerns which should be addressed.

Concerns

Major: 1. Line 142: Please comment on whether analyzing only 1 untreated HER2 positive tumor for which multi-region samples from primary breast tumors was available would be enough to understand the potential impact of ITH in HER2 targeted treatment induced genomic changes, and to draw the conclusions. A few additional cases of untreated Her2 positive primary breast cancer would add a lot of strength to the conclusions.

We have identified two new cohorts of multi-region sampled untreated primary breast tumors published in 2018 (Ullah et al, *J Clin Invest*, 2018, PMID 29480816 and Barry et al, *Clin Cancer Res*, 2018, PMID 29891724) that we now include in our analysis. From these cohorts, we have added to our analyses the cases (one HER2+) that meet our quality thresholds (purity > 40% in each analyzed region, and at least 40 mutations with coverage of at least 20 reads across all regions), bringing the total number of untreated tumors included in Figure 1 to 15. The new untreated HER2+ tumor is from the Ullah et al cohort, and has *HFR* 0.135 and *Fst* 0.240.

While we agree that additional cases of untreated multi-region sequenced HER2+ breast cancer would be desirable to support the observation that this subtype appears similar (with respect to between-region genetic divergence) to the other, more represented subtypes, we believe that our findings are robust with the two that are available. We note that we do not find substantial differences between ER+/HER2- tumors (N=6, mean *HFR* 0.35) and TNBC tumors (N=7, mean *HFR* 0.20), which are generally thought to be more distinct from each other than either is from HER2+ subtypes, which can span multiple PAM50 classifications (Prat et al, *Oncotarget*, 2017 PMID 29088709). We are also reassured that the treated, HER2+ tumors we analyzed exhibited

comparable *HFR* to the untreated cohort that was more heavily ER+/HER2- or TNBC. As the reviewer noted in the point below, these treated tumors did not have major responses to neoadjuvant therapy, and the three in particular that did not undergo clonal replacement might be expected to resemble pre-treatment HER2+ tumors in between-region genetic divergence. Furthermore, we might have hypothesized that heterogeneity would decrease with clonal replacement, as observed in two of these tumors, and yet we see high heterogeneity even so. Overall, based on our results, we infer that it is very unlikely that untreated HER2+ tumors have substantially lower heterogeneity than the other subtypes examined.

Major: 3. Line 212: *HFR* was similar between multi-region sampled pre- (30%, range 1-70%) and post- 213 (28%, range 10-54%) treatment tumors “: Please comment whether this is perhaps expected as the analyzed tumors were almost all pT3 tumors and did not respond to neoadjuvant chemotherapy, so similar *HFR* is expected?

We agree it is important to clarify that heterogeneity might change differently in tumors with greater treatment responses. We have changed the subtitle of this section of the Results and its text to account for the fact that the tumors we analyzed had bulky residual disease (page 6) and have noted in the Discussion (page 11) that the impact of treatment on heterogeneity for tumors with more favorable results is currently unknown. At the same time, we note that we had hypothesized that between-region heterogeneity would change in tumors that undergo drastic tumor evolution in the form of a clonal sweep (seen in P1 and P5). The *HFR* values for P1 (0.26) and P5 (0.11) fell in the same range as the *HFR* values for the other treated tumors that did not undergo a clonal sweep, potentially suggesting that the well of heterogeneity runs deep, such that even after a clone sweeps through, heterogeneity remains relatively high.

Major: Line 262-264” and observed that the degree of clonal change seen in P5 was statistically unlikely to have resulted from pre-existing heterogeneity (*tHFR* > 99th percentile). In the previous section Line 249-250 the argument was made that taking only 1 biopsy ; 30% (P5: *tHFR* = 0.3 > 90th percentile) of the clonal mutations would be absent in the other biopsy. Maybe I don’t understand, but how do these analyses relate to each other? Please comment.

We thank the reviewer for bringing up this point of confusion. In this paper, we define two related, but distinct, statistics of clonal change: *HFR* relates to geographic clonal change, while *tHFR* relates to temporal clonal change. The first (*HFR*) can be translated as the reviewer does here: i.e. *HFR* = 0.3 means that 30% of the clonal mutations would be rare or absent in the other biopsy. For P5, the *HFR* after treatment is actually 0.11 (i.e. 11% of the clonal mutations discovered post-treatment would be rare or absent in another post-treatment region) (these values were discussed in the previous section of the Results). “*tHFR*” (or “temporal” *HFR*) has a different purpose; here we are looking at the degree of change across time, rather than across space, and so compare all of the biopsies from one time (here, the pre-treatment time, where we always have only one such biopsy) to all of the biopsies from another time (here, the post-treatment time, where we have two or more biopsies). A *tHFR* value calculated with one pre-treatment and one post-treatment biopsy would reflect both the regional variation in *HFR* and any clonal change across time; thus we see that the “one post-treatment sample” simulated distribution of the null case (no change across treatment) shown in Figure 3D corresponds closely to the simulated distribution of *HFR* shown in Figure 1C. With multi-region sampling at least one time point, *tHFR* is able to disentangle (and measure) clonal change across time from clonal change across space. The *tHFR* value of 0.3 for P5 thus represents the degree of clonal change from pre- to post-treatment; i.e. *tHFR* = 0.3 means that 30% of the clonal mutations present in all post-treatment regions were absent or rare in the pre-treatment region. As the reviewer notes, the value of 0.3 is quite similar to the geographic heterogeneity (*HFR*) seen in some breast tumors, which is why multi-region sampling post-treatment is necessary to distinguish geographic heterogeneity from clonal change across treatment. (If the heterogeneity across treatment with true clonal change were always greater than that within a timepoint, we

could simply detect it with one pre- and one post-treatment region, as to some extent we can in P1.)

We have clarified the distinction between *HFR* and *tHFR*, and what each measures, in the Results section where noted by the reviewer (page 7).

Major: Line 73-76 “Such an evolutionary model would have implications for treatment, including whether adjuvant therapies should be targeted based on the genetic composition of the post-treatment rather than the pre-treatment tumor.” And Line 382 and 384: “The frequency with which ... adjuvant therapy”

I guess that the authors are implying here the setting of adjuvant treatment which if given in patients treated with neo-adjuvant treatment and surgery? By inquiring our medical oncologists adjuvant treatment decisions in breast cancer are already made based on pre- and post treatment tumor characteristics. This treatment options and decisions are made during multi-disciplinary meetings where surgeons, medical oncologist, pathologists and radiologists discuss breast cancer patients. Please comment.

This is interesting to know; at our primary institution (Stanford), re-testing specimens after neoadjuvant therapy is not standard, and there are currently no pathology guidelines recommending re-testing on anything but a metastatic specimen. We are not surprised, though, that there is variability in practice on this question, as it is an active area of research whether there would be any clinical benefit in re-testing specimens after neoadjuvant therapy to guide treatment. As an example, one report on standardized pathological characterization of residual disease in neoadjuvant clinical trials (Bossuyt et al, 2015, *Annals of Oncology*, PMID 26019189) notes that “in current practice, the choice of adjuvant therapy is dictated by the results at primary diagnosis”, but both in this report and in other studies (e.g. Xian et al, 2017, *Human Pathology*, PMID 28041972), it is clear that clarification of the potential role of re-testing is needed. We cannot comment on this issue as regards ER/PR/HER2 testing, but our hope is to provide some insight into its potential importance as we enter an era of increased molecular profiling of tumor specimens, including whole-exome/targeted sequencing. In other words, our results indicate that there can, in a subset of cases, be massive shifts in clonal mutations with treatment, and our hope is to motivate multi-region analysis of the surgical specimen going forward in any future studies that may explore genomic biomarker-targeted adjuvant therapy (whether or not preceded by neoadjuvant therapy). We have added the two above references outlining the ongoing research into this area to the Introduction at the sentence noted by the reviewer (page 3), and have added clarification to the Discussion section (page 10) that our results specifically provide insight into the question of when and how to perform tumor profiling to determine adjuvant therapy guided by potential novel genomic (mutational or copy number) biomarkers.

Minor: 2. Line 202: It is difficult to follow how many of the 20 primary Her2 positive breast biopsies (pre-treatment) had interpretable bulk whole-exome sequencing. Please comment / add to the manuscript or add to figure suppl 5 whether the drop out of 45%, was only due to low cellularity at time of surgery or due to low quality DNA of the pre-treatment biopsy or low cellularity of the pre-treatment biopsy.

We did not include pre-treatment biopsy cellularity as an exclusionary criterion in Figure S5 because, as it turned out, there were no tumors that we needed to exclude solely based on pre-treatment biopsy cellularity. However, of the 9 tumors we excluded based on poor post-treatment cellularity (as measured in one of two ways as outlined in Figure S5), 2 would also have been excluded based on poor pre-treatment cellularity (the other 7 had adequate

cellularity pre-treatment but not post-treatment). In other words, the reviewer is correct in pointing out that while most tumor exclusions were due to a post-treatment cellularity issue, there were cases where the tumor was not analyzable due to low cellularity at time of diagnosis (even without the effects of treatment). We have added this point to the legend of Figure S5.

Minor: Line / Paragraph 268-277: This might be a difficult paragraph to follow for the reader. What is the added value of this paragraph if there is only a trend and not significant t-test between the pre- and post treatment samples of the tumors that have undergone clonal replacement compared to tumors that didn't undergo a clonal replacement. Please comment. Might help to add MEDICC53 ref in this paragraph.

Undoubtedly, copy number clonal change is more complicated to assess than mutational clonal change. One of the reasons is that it is quite difficult to assess subclonal copy number, especially in the face of fluctuating purity, and to disentangle various copy number-altered subclones. Another is simply that it is more difficult to accurately call copy number from whole-exome sequencing data than it is to call somatic mutations. Nonetheless, methods exist (like MEDICC which we used, and have now cited here per the reviewer's recommendation) to assess copy number distance in between samples. We felt it was important to apply these methods to determine whether copy number data supported our findings from the mutational data. We favor keeping this analysis, recognizing its limitations, as it provides some orthogonal support for our findings from the mutational data, and because it introduces methodology for assessing clonal copy number change between regions and across time, which we believe should be undertaken in any such analysis.

Minor: LINE / Paragraph 279-292: The analyses in this paragraph are completely underpowered to make any conclusions about potential causes of resistance to HER2-targeted therapy combined with chemotherapy. Please comment and how do these results relate to earlier published results?

We agree. We did want to provide a Supplementary Table including the genetic alterations unique to the two resistant clones that we identified, as they represent candidate drivers of resistance that have not been previously reported, but we concur that we are absolutely underpowered to make any comment on which of these alterations might be the actual driver. This method of identifying putative resistance drivers (by examining alterations present in a clone that has swept through after treatment) is different from most previous methods to identify drivers of resistance in patient cohorts, which examined only baseline alterations (e.g. PMIDs 20813970, 23650412, 25199759, 25559818). Similarly, in a recent paper Kim et al (*Cell*, 2018, PMID 29681456) reported the set of copy number changes present in the clones they sequenced from single cells post-treatment. Given the reasonable prevalence of clonal replacement as we suggest here (~10%), it might be a tractable approach to examine multi-region sequencing of large numbers of patients before and after treatment to gain power to search for drivers of resistance in this way, especially if such patients were already undergoing multi-region sequencing to help determine targeted therapy in future precision oncology efforts.

As we agree with the reviewer that these findings are hypothesis-generating only, and we do not wish to bog down readers, we have replaced this paragraph with a one-sentence point that the genetic alterations in resistant clones are candidate resistance drivers, and that we provide them in the Supplementary Tables 5 and 6.

REVIEWERS' COMMENTS:

Reviewer #1 (Remarks to the Author):

The authors have addressed successfully all of my comments. They have improved substantially the clarity of the manuscript. I do not have any additional remark.